# Serotonergic and Adrenergic Neuroreceptor Manipulation Ameliorates Core Symptoms of ADHD through Modulating Dopaminergic Receptors in Spontaneously Hypertensive Rats

**DOI:** 10.3390/ijms25042300

**Published:** 2024-02-15

**Authors:** Sampath Madhyastha, Muddanna S. Rao, Waleed M. Renno

**Affiliations:** Department of Anatomy, College of Medicine, Kuwait University, Safat 13110, Kuwait; muddanna.rao@ku.edu.kw (M.S.R.); waleed.renno@ku.edu.kw (W.M.R.)

**Keywords:** ADHD, alpha-2 adrenergic receptor, attention, impulsivity, DA-D1, DA-D2, 5-HT1A, 5-HT2A, prefrontal cortex, striatum and substantia nigra

## Abstract

The core symptoms of attention deficit hyperactivity disorder (ADHD) are due to the hypofunction of the brain’s adrenergic (NE) and dopamine (DA) systems. Drugs that enhance DA and NE neurotransmission in the brain by blocking their transporters or receptors are the current therapeutic strategies. Of late, the emerging results point out the serotonergic (5-HT) system, which indirectly modulates the DA activity in reducing the core symptoms of ADHD. On this basis, second-generation antipsychotics, which utilize 5-HT receptors, were prescribed to children with ADHD. However, it is not clear how serotonergic receptors modulate the DA activity to minimize the symptoms of ADHD. The present study investigates the efficacy of serotonergic and alpha-2 adrenergic receptor manipulation in tackling the core symptoms of ADHD and how it affects the DA neuroreceptors in the brain regions involved in ADHD. Fifteen-day-old male spontaneously hypertensive rats (SHRs) received 5-HT1A agonist (ipsapirone) or 5-HT2A antagonist (MDL 100907) (i.p.) or alpha-2 agonist (GFC) from postnatal days 15 to 42 along with age-matched Wistar Kyoto rats (WKY) (n = 8 in each group). ADHD-like behaviors were assessed using a battery of behavioral tests during postnatal days 44 to 65. After the behavioral tests, rat brains were processed to estimate the density of 5-HT1A, 5-HT2A, DA-D1, and DA-D2 neuroreceptors in the prefrontal cortex, the striatum, and the substantia nigra. All three neuroreceptor manipulations were able to minimize the core symptoms of ADHD in SHRs. The positive effect was mainly associated with the upregulation of 5-HT2A receptors in all three areas investigated, while 5-HT1A was in the prefrontal cortex and the substantia nigra. Further, the DA-D1 receptor expression was downregulated by all three neuroreceptor manipulations except for alpha-2 adrenergic receptor agonists in the striatum and 5-HT2A antagonists in the substantia nigra. The DA-D2 expression was upregulated in the striatum while downregulated in the prefrontal cortex and the substantia nigra. In this animal model study, the 5-HT1A agonist or 5-HT2A antagonist monotherapies were able to curtail the ADHD symptoms by differential expression of DA receptors in different regions of the brain.

## 1. Introduction

Attention deficit hyperactivity disorder (ADHD), a complex heterogeneous neural illness, disturbs a huge global population of children and adolescents. Approximately 11% (6.4 million) of children aged between 4 and 17 years were detected as having ADHD in the United States as of 2014 [1]. The etiology of this neurodevelopmental disorder is diverse. It includes gestational, perinatal, and environmental factors and the presence of variants of some genes. Further, the manifestations of ADHD are highly variable among children and are classified into three subtypes depending on the intensity of the symptoms: predominantly inattentive, predominantly hyperactive–impulsive, and combined. Behavioral intervention is the first-line treatment, but most children require pharmacological treatment. Due to the heterogeneous etiology and diverse manifestations of ADHD, a personalized treatment is required to obtain the optimum effect. This complex neurodevelopmental disorder mainly involves imbalance in the neurotransmitters dopamine (DA) and noradrenaline (NE) in certain regions of the brain [2]. Apart from these two neurotransmitters, serotonin (5-HT) is also known to be involved in ADHD [3]. Serotonin is implicated in inhibitory response control [4], and reduced 5-HT levels have been associated with a reduction in sustained attention. The pharmacological basis of treating ADHD core symptoms like inattentiveness, hyperactivity, and impulsivity is enhancing the NE, DA, and 5-HT levels in extraneuronal space either by blocking their reuptake, by blocking their receptors, or by blocking their transporters.

The available pharmacological treatment options include stimulants and second-generation antipsychotics. The stimulant medication as first-line pharmacotherapy for ADHD includes methylphenidate and amphetamine [5]. The mechanism of action of methylphenidate is by inhibiting DA and NE transporters, thereby increasing the level of DA and NE [6] in the extraneuronal space, especially in the prefrontal cortex, which regulates attention behavior. Further, methylphenidate has agonist activity on 5-HT1A neuroreceptors [6]. A large number of data suggest the efficacy of methylphenidate [7,8] with minor and generally acceptable adverse effects. Contrary to this, Childress and Sallee [9] report that about 20–35% of patients show inadequate response to stimulants. However, methylphenidate addiction and dependence remain controversial to date. The U.S. Drug Enforcement Administration (DEA) classifies methylphenidate as a Schedule II substance. The second-generation antipsychotics or non-stimulants for ADHD mainly include atomoxetine, quetiapine, and risperidone. Stimulants enhance dopamine levels between neurons, while antipsychotics act by blocking their effects on dopamine receptors.

It is known that stimulant and antipsychotic medications have opposing mechanisms of action. Stimulants are known to enhance the DA and NE in the synaptic cleft (extraneuronal space), while antipsychotics work by blocking their effects through their receptors. Though the focal therapeutic sites of stimulants and antipsychotics are different, they both interact at the same receptor subtypes and do so in the same parts of the brain [10]. Examination of DA pathways has revealed that stimulants have significant activation at both the limbic system and the cortex [11], while antipsychotics have their strongest effects in the limbic system [12], but they also have effects on the cortex [13,14]. Hence, the concurrent usage of stimulant and antipsychotic use has been rationalized by suggesting that they likely interact with different receptor subtypes and do so in different pathways of the brain. Atomoxetine acts by selectively blocking the NE transporter in the prefrontal cortex to enhance NE and DA levels [15]. Atomoxetine is considered a first-line therapy for patients at risk for substance abuse disorders [16]. Several studies indicate the efficacy of atomoxetine in minimizing the core symptoms of ADHD, like inattention, impulsivity, and hyperactivity [17,18]. The most common adverse effects of atomoxetine in children with ADHD include gastrointestinal symptoms, sleep disturbances, cardiovascular adverse reactions, irritability, dizziness, fatigue, and headache. The significant side effects of atomoxetine include elevated liver enzymes [19] and motor tics [20,21]. Analyses of twelve placebo-controlled trials in children and adolescents showed a greater risk of suicidal ideation during treatment with atomoxetine [22]. Apart from atomoxetine, clonidine and guanfacine are also used as non-stimulant medications for ADHD. They act as agonists to alpha-2A neuroreceptors, especially in the prefrontal cortex, to enhance the NE neurotransmission. Their efficacy in dealing with ADHD is also widely described [23,24]. As alpha-2 agonists are antihypertensive agents, blood pressure and heart rate should be monitored routinely, and they should not be discontinued abruptly to prevent a hypertensive crisis. The known adverse effects of alpha-2A agonists include somnolence, fatigue, irritability, insomnia, dry mouth, sedation, and rarely bradycardia and syncope. There was a case report with four cases of sudden death in children aged between 8 and 10 years taking a combination of clonidine and methylphenidate [25]. Then comes risperidone, which has a wide range of pharmacological action compared to atomoxetine or stimulant medications. Risperidone has an affinity for DA-D2, 5-HT2A, alpha-1, and alpha-2 adrenergic and H1 receptors. Its efficacy in reducing ADHD symptoms was attributed to its ability to block the dopamine 2 (D2) receptors, specifically in the mesolimbic pathway [26]. Both methylphenidate and atomoxetine do not directly involve dopamine and 5-HT receptors. Though risperidone is widely used in schizophrenia and autism spectrum disorders, it is also found to be effective and safe in treating children and adolescents with ADHD [27]. Risperidone therapy is known to cause inappropriate speech retardation and more frequently extrapyramidal side effects [28]. The other known adverse effects include weight gain, prolonged QT interval, tardive and withdrawal dyskinesia, diabetes mellitus, and hyperlipidemia [29,30,31]. Being a relatively new generation of medication, its potential benefits and adverse effects in long-term use and comorbid conditions are yet to be explored. Quetiapine is an atypical antipsychotic used in treating schizophrenia, bipolar disorder, and depression [32]. Quetiapine promotes an increase in prefrontal dopamine release through the antagonism of 5-HT2A receptors and partial agonism of 5-HT1A receptors [33]. Quetiapine is more specific in its action, which acts as a DA-D2 neuroreceptor antagonist in the mesolimbic pathway and also as an antagonist to 5-HT2A neuroreceptors in the frontal cortex [34]. Quetiapine has been shown to be effective in reducing ADHD symptoms and aggression [35,36].

From these results, it is understood that 5-HT2A antagonism and 5-HT1A agonism are required in minimizing the hyperactivity behavior of ADHD. A further alpha-2 adrenergic receptor agonist is also a therapeutic combination in these second-generation antipsychotics. Hence, in the present study we considered these three neuroreceptor manipulations. The 5-HT system has been implicated in the pathophysiology of ADHD [37,38], but its specific clinical relevance, and in particular its relevance for novel therapeutic strategies, remains to be investigated. The available reports on the specific role of 5-HT1A and 5-HT2A in ADHD are highly contradictory. Stimulation of 5-HT1A receptors has been associated with reducing anxiety and reducing depression. Chronic treatment with 5-HT1A receptor agonists (for example, buspirone) has been known to cause antidepressant and anxiolytic effects in both animal and human studies. In contrast to this, Riley and Overton [39] suggest utilizing 5-HT1A antagonists as a potential way forward in the development of pharmacotherapies for ADHD.

There are contradictory reports regarding the use of 5-HT2A antagonists in pharmacotherapy directed towards ADHD. For instance, decreased 5-HT2A neuroreceptors are associated with enhanced attention and working memory while their increased expression in the prefrontal cortex is associated with decreased inhibitory response and reversal learning [40]. Contrary to this, a 5-HT2A antagonist resulted in impairment of sustained attention and motor impulse control [41]. Further, the role of 5-HT2A in the prefrontal cortex in impulsive response was demonstrated by many authors [42,43] where stereotaxic injection of 5-HT2A agonists into the medial prefrontal cortex resulted in increased impulsive responses. Though it is understood that each of these neuroreceptors functions differently in different regions of the brain, the role of 5-HT1A and 5-HT2A needs to be investigated further. Further, the concurrent usage of dopamine receptors in antipsychotics to treat ADHD is of concern because the subvariety of dopamine receptors has different roles in different parts of the brain. To date, there are no studies showing the effect of serotonergic receptor manipulation’s effects on expression of DA receptors. Previously, the selective serotonin reuptake inhibitor fluoxetine reduced the DA content in the striatum [44] but enhanced DA content in the prefrontal cortex [45]. This may be due to the diverse roles of dopamine receptors in the prefrontal cortex and the striatum. There is a general consensus that 5-HT2A activation stimulates DA release in the ventral tegmental area or 5-HT2A acts to facilitate ventral tegmental neuronal activity and DA release [46]. The currently approved DA agonists are mainly D1/D2/D3 agonists which produce side effects typical of D2/D3 agonists, namely a tendency to increase addictive behaviors like gambling, compulsive shopping, and hypersexuality. Long-term use of DA agonists may cause choreiform and dystonic movements and psychiatric disturbances. Hallucinations, delusions, confusion, depression, and mania are some of the most common adverse effects related to the long-term use of dopamine agonists. Hence, formulation of DA agonists as a part of ADHD medication for children raises concerns for the possible extrapyramidal symptoms. Drugs most closely related to DA are very quickly metabolized (because they are catechol derivatives, potentially toxic), so that the plasma concentration required to affect the brain gives unacceptable side effects like hypotension. With this background, the present study is designed to evaluate the efficacy of serotonergic receptor manipulation on minimizing the core symptoms of ADHD and to test how dopamine receptors are affected in the prefrontal cortex, striatum, and the substantia nigra. Since alpha-2 adrenergic agonists are well-known therapeutic components of ADHD treatment, the experiment also included an animal group who received alpha-2 adrenergic agonists. Alpha-2A agonist guanfacine hydrochloride (GFC, Cat. No. 1030) was administered (0.3 mg/kg during week one, 0.45 mg/kg during week two, and 0.6 mg/kg during week three) intraperitoneally.

A detailed estimation of these neuroreceptor expressions in the areas of the brain involved in ADHD during the treatment phase provides more insight into the correlation and interaction between the brain amines during the therapy phase. Understanding the relationship of dopamine receptors with ADHD will help us to elucidate different roles of these receptors and to develop therapeutic approaches for ADHD. The present study is likely to provide more insight into the different roles of different receptors in different parts of the brain in alleviating the symptoms of ADHD. Further, the link, if any, between serotonergic and DA systems can be revealed. The results of the study are likely to throw more insights into the role of these neuroreceptors in the brain regions concerned with ADHD.

## 2. Results

### 2.1. General Locomotor and Exploratory Behavior

The mean distance traveled, and mean movement velocity were significantly high (*p* < 0.001) in SHRs who did not receive any treatment (SHR-NC) compared to their counterparts, WKY rats who also did not receive any treatment (WKY-NC). It indicates that at baseline SHRs are hyperactive compared to their counterpart WKY rats. SHRs (ADHD model) who received either MDL (5-HT2A antagonist) or IPS (5-HT1A agonist) or GFC (alpha-2 adrenergic agonist) showed a highly significant (*p* < 0.001) decrease in these study parameters (mean distance traveled, mean movement velocity) when compared to the SHR-NC group (Figure 1A,B). In order to simplify the findings and to avoid distraction, behavioral outcomes of the WKY-MDL, WKY-IPS, and WKY-GFC groups are not shown in all graphs and video tracking as their statistical comparison was similar to the SHR treated group.

The mean distance traveled, and time spent in the peripheral zone of the open field apparatus were significantly (*p* < 0.001) high in control SHRs when compared to their counterpart WKY control rats. Treatment of MDL or IPS or GFC in SHRs significantly reduced (*p* < 0.001) these study parameters when compared with control SHRs (Figure 1C,D). There was no significant difference observed between MDL and IPS groups. Representative video tracking is shown in Figure 1E. The anxiety-like behavior expressed by SHRs by spending more time in the peripheral zone of the apparatus is minimized by the drugs used in this study. The beneficial effect was associated with upregulation of 5-HT2A receptors. Further, the DA1 receptor expression was downregulated by all three neuroreceptor manipulations except for alpha-2 adrenergic receptor agonist in the striatum and 5-HT2A antagonists in the substantia nigra. DA-D2 expression was upregulated in striatum as well as substantia nigra while downregulated in the prefrontal cortex in all three neuroreceptor manipulations.

### 2.2. Locomotor Hyperactivity

The mean distance traveled and mean score of movement velocity were significantly high (*p* < 0.01) in control SHRs compared to their counterpart WKY control rats when measured for a long duration of 1 h. This clearly demonstrates locomotor hyperactivities in SHRs compared to their counterpart WKY rats. SHRs who received either MDL or IPS or GFC showed a highly significant (*p* < 0.01) decrease in mean score of movement velocity when compared to control SHRs in the study (Figure 2A,B), indicating a beneficial effect of these neuroreceptor manipulations in curtailing the hyperactivity.

The mean number of movements and movement time were significantly (*p* < 0.01) less in WKY control rats when compared to control SHRs. SHRs who received either MDL or IPS or GFC showed a highly significant (*p* < 0.01) decrease in mean distance traveled and mean score of velocity when compared to control SHRs (Figure 2C,D). There was no significant difference noticed between MDL and IPS groups in these parameters. This indicates that both alpha-2 adrenergic agonist and 5-HT2A antagonist had similar effects in reducing the locomotor hyperactivities. Representative video tracking is shown in Figure 2E. The hyperactivity by means of excessive movements shown by SHRs is suppressed by both the drugs used in this study.

### 2.3. Anxiety Level

The percentage of time spent in the open arm of an elevated plus maze was significantly (*p* < 0.001) less in control SHRs when compared with WKY control rats. The percentage of time spent in the closed arm of the test apparatus was significantly less in WKY control rats compared to control SHRs, indicating higher anxiety in SHRs. SHRs who received either MDL or IPS or GFC showed a significant (*p* < 0.001) decrease in the percentage of time spent in the closed arm of the elevated plus maze (Figure 3A). This demonstrates that all three neuroreceptor manipulations were able to reduce the anxiety in SHRs. There was no significant difference between MDL- and IPS-treated groups with regard to these parameters. This indicates that both the 5-HT1A agonist and 5-HT2A antagonist have the same degree of effect in reducing the anxiety in SHRs. Representative video tracking is shown in Figure 3B. This indicates that the anxiety-like behavior expressed by SHRs is suppressed by treatment with these drugs.

### 2.4. Impulsivity (Tolerance Delay) in Modified T-Maze Test

As expected, there was a progressive increase in the choice of the large-reward arm from day 1 to day 3 during the learning phase in all rat groups. During test sessions, the control SHRs showed significantly (*p* < 0.001) fewer choices of the large, delayed reward arm compared to WKY control rats. Treatment with MDL, IPS, or GFC significantly (*p* < 0.001) increased the choice of the large, delayed reward arm when compared to control SHRs (Figure 4A).

The mean percentage choice of the large but delayed reward arm was significantly (*p* < 0.001) high in WKY control rats compared to control SHRs during test sessions. Treatment with MDL or IPS or GFC in SHRs resulted in a significant (*p* < 0.001) increase in the percentage of choice of the large but delayed reward arm (Figure 4B). However, there was no significant difference between MDL- and IPS-treated groups with regard to these parameters. The result of this study indicates that the impulsivity and disinhibition behavior expressed by SHRs are reversed by drug treatment.

### 2.5. Impulsive Water Drinking (in Aversive Electro Foot Shock Apparatus)

During test sessions, the control SHRs showed a significantly (*p* < 0.001) higher frequency of entry into the water area for drinking water despite the foot shock compared to WKY control rats (Figure 5A). Treatment with MDL or IPS or GFC significantly (*p* < 0.001) reduced the number of entries into the water area compared to the control SHRs (Figure 5B). There was no significant difference between MDL and IPS groups with regard to these parameters. Representative video tracking is shown in Figure 5C. The impulsivity shown by SHRs was minimized by drug treatment.

### 2.6. Neuroreceptor Quantity and Expression in Prefrontal Cortex, Striatum, and Substantia Nigra

Expression of neuroreceptors (5-HT2A, 5-HT1A, DA-D1, and DA-D2) in prefrontal cortex, striatum, and the substantia nigra was measured in the tissue lysate by the Western blot method using appropriate antibodies against the receptors. For Western blot analysis, we included three additional WKY groups treated with receptor agonists and antagonists (WKY-MDL, WKY-IPS, and WKY-GFC). We present below the immunoblots of all the groups and quantified data of six selected groups for convenient comparison.

In the prefrontal cortical region, the medial portion showed higher alteration in receptor density compared to lateral or orbital regions. In the striatum, the receptor alteration was more uniform in the ventral and dorsal parts. All these regions of the brain, which are involved in ADHD, showed inconsistent expression of neuroreceptors, especially DA receptors, possibly indicating their diverse roles in different parts of the brain. The implications of these results are discussed in the Discussion section.

#### 2.6.1. Prefrontal Cortex

5-HT2A neuroreceptors: At baseline, though there was slight reduction in 5-HT2A receptor density in SHRs compared to WKY rats in the prefrontal cortex, it was statistically not significant (*p* > 0.05). However, receptor protein level was significantly increased in all SHR treated groups compared with the SHR-NC group (SHR-NC vs. SHR-MDL, *p* < 0.05; SHR-NC vs. SHR-IPS, *p* < 0.05; SHR-NC vs. SHR-GFC, *p* < 0.05; Figure 6A,B and Figure 7).

5-HT1A neuroreceptors: At baseline, there was a slight reduction in the expression of 5-HT1A receptor protein level when compared between SHR-NC and WKY-NC, but this was statistically not significant (*p* > 0.05). However, receptor protein level was significantly increased in IPS-treated (5-HT1A agonist, SHR-NC vs. SHR-IPS, *p* < 0.05) and GFC-treated (alpha-2A agonist, SHR-NC vs. SHR-GFC, *p* < 0.05) SHR groups compared with SHR-NC, but not in that treated with MDL (5-HT2 antagonist) (SHR-NC vs. SHR-MDL, *p* > 0.05, Figure 6A,C and Figure 7).

DA-D1 neuroreceptors: At baseline, the DA-D1 neuroreceptor protein level did not differ between SHR-NC vs. WKY-NC (*p* > 0.05). The DA-D1 receptor protein level was significantly reduced in all the treated groups compared with SHR-NC (SHR-NC vs. SHR-MDL, *p* < 0.05; SHR-NC vs. SHR-IPS, *p* < 0.05; SHR-NC vs. SHR-GFC, *p* < 0.05; Figure 6A,D and Figure 8).

DA-D2 neuroreceptors: The DA-D2 neuroreceptor protein level did not differ between SHR-NC vs. WKY-NC (*p* > 0.05) at baseline. The DA-D2 receptor protein level was significantly reduced in all three treated groups (5-HT1A agonist, SHR-NC vs. SHR-IPS, *p* < 0.05; alpha-2A agonist, SHR-NC vs. SHR-GFC, *p* < 0.05; 5-HT2A antagonist, SHR-NC vs. SHR-MDL, *p* < 0.05) (Figure 6A,E and Figure 8). Immunostaining for the above receptors confirmed the expression of the receptors in the prefrontal cortical sections. Expression of the above receptor proteins is comparable to data obtained from Western blot analysis for the receptors (Figure 7 and Figure 8).

#### 2.6.2. Striatum

5-HT2A neuroreceptors: In the striatum, there were no significant changes in the 5-HT2A receptor protein level compared between SHR-NC and WKY-NC (*p* > 0.05) at the baseline. However, 5-HT2A receptor protein level was significantly increased in IPS- and MDL-treated groups compared with the SHR-NC group (5-HT1A agonist, SHR-NC vs. SHR-IPS, *p* < 0.05; 5-HT2A antagonist, SHR-NC vs. SHR-MDL *p* < 0.05). However, the 5-HT2A expression was significantly reduced by GFC treatment of alpha-2 adrenergic agonist, SHR-NC vs. SHR-GFC, *p* < 0.05; Figure 9A,B and Figure 10.

5-HT1A neuroreceptors: There were no significant changes in the 5-HT1A receptor protein level when compared between SHR-NC and WKY-NC (*p* > 0.05) at the baseline. However, receptor protein level was significantly decreased in MDL-treated (5-HT2A antagonist, SHR-NC vs. SHR-MDL, *p* < 0.05) and IPS-treated (5-HT1A agonist, SHR-NC vs. SHR-IPS, *p* < 0.05) groups compared with the SHR-NC group. The 5-HT1A receptor level was increased in the SHR-GFC (alpha-2A agonist) group compared with SHR-NC (SHR-NC vs. SHR-GFC, *p* < 0.05; Figure 9B,C and Figure 10).

DA-D1 neuroreceptors: The DA-D1 receptor protein level expression did not differ between SHR-NC vs. WKY-NC at the baseline (*p* > 0.05). The DA-D1 receptor protein level was reduced in the SHR group treated with IPS (5-HT1A agonist, SHR-NC vs. SHR-IPS, *p* < 0.05) and MDL (5-HT2A antagonist, SHR-NC vs. SHR-MDL, *p* < 0.05). However, the SHRs treated with GFC (alpha-2A agonist) showed an increased expression of DA-D1 receptors compared to the SHR control group (alpha-2A agonist, SHR-NC vs. SHR-GFC, *p* < 0.05; Figure 9A,D and Figure 11).

DA-D2 neuroreceptors: At baseline, the SHR-NC group did not show any significant difference in expression of DA-D2 receptors when compared with WKY-NC (*p* > 0.05). The SHR groups treated with IPS (5-HT1A agonist, SHR-NC vs. SHR-IPS *p* < 0.05) or GFC (alpha-2A agonist, SHR-NC vs. SHR-GFC *p* < 0.05) or MDL (SHR-NC vs. SHR-MDL, *p* < 0.05) showed a significant increase in DA-D2 receptor expression compared to SHR-NC (Figure 9A,E and Figure 11).

Immunostaining for the above receptors confirmed the expression of the receptors in the dorsal striatum sections. Expression of the above receptor proteins is comparable to data obtained from Western blot analysis for the receptors (Figure 10 and Figure 11).

### 2.7. Substantia Nigra

5-HT2A neuroreceptors: Like in the prefrontal cortex, there were no significant changes in the 5-HT2A receptor protein level in the substantia nigra when compared between SHR-NC and WKY-NC (*p* > 0.05). However, 5-HT2A receptor protein level was significantly increased in all three treated groups when their value was compared with the SHR-NC group (SHR-NC vs. SHR-MDL, *p* < 0.05; SHR-NC vs. SHR-IPS, *p* < 0.05; SHR-NC vs. SHR-GFC, *p* < 0.05; Figure 12A,B and Figure 13).

5-HT1A neuroreceptors: There were no significant changes in the 5-HT1A receptor protein level when compared between SHR-NC and WKY-NC (*p* > 0.05) at the baseline. However, receptor protein level was significantly increased in MDL-treated (5-HT2A agonist), IPS-treated (5-HT1A agonist) as well as GFC-treated (alpha-2A agonist) SHR groups compared with SHR-NC (SHR-NC vs. SHR-MDL, *p* < 0.05; SHR-NC vs. SHR-IPS, *p* < 0.05; SHR-NC vs. SHR-GFC, *p* < 0.05; Figure 12A,C and Figure 13).

DA-D1 neuroreceptors: At baseline, there was no difference between SHR-NC and WKY-NC (*p* > 0.05) with regard to DA-D1 receptor expression. SHR groups treated with IPS (5-HT1A agonist) or GFC (alpha-2A agonist) had a significantly reduced expression of DA-D1 neuroreceptor protein compared to SHR-NC (SHR-NC vs. SHR-IPS, *p* < 0.05, SHR-NC vs. SHR-GFC, *p* < 0.05). However, the SHR-MDL (5-HT-2A antagonist) therapy resulted in a significant increase in DA-D1 receptor protein compared to SHR-NC (SHR-NC vs. SHR-MDL *p* < 0.05; Figure 12A,D and Figure 14).

DA-D2 neuroreceptors: The DA-D2 receptor protein level was significantly reduced in all three treatment groups when compared with the SHR-NC group (5-HTA agonist, SHR-NC vs. SHR-IPS, *p* < 0.05; SHR-NC vs. SHR-MDL, *p* < 0.05; SHR-NC vs. SHR-GFC, *p* < 0.05; Figure 12A,D and Figure 14).

Immunostaining for the above receptors confirmed the expression of the receptors in the substantia nigra sections. Expression of the above receptor proteins is comparable to data obtained from Western blot analysis for the receptors (Figure 13 and Figure 14).

## 3. Discussion

The spontaneously hypertensive rats (SHRs) showed most of the characteristic behaviors of ADHD like locomotor hyperactivity (open field test results), anxiety-like behavior (elevated plus maze results), impulsivity, and inattention (modified T-maze and water drinking test results) compared with their counterpart WKY rats. This confirms the effectiveness of SHRs as an animal model for ADHD. Both 5-HT1A agonist and 5-HT2A antagonist were able to curtail ADHD-like behavior on par with alpha-2 adrenergic agonist treatment which is well known to minimize ADHD symptoms. These findings showed that the neuroreceptor manipulation attempted in this study helped to reduce the hyperactive and attention deficit behaviors in SHRs. It has been well established that reduced levels of NE and DA in the brain are associated with ADHD. Of late, it has also been shown that a reduced 5-HT level in the prefrontal cortex is a characteristic of ADHD-like impulsive and hyperactive behavior which could be reversed with selective 5-HT reuptake inhibitors [39]. Though it is understood that 5-HT plays a major modulatory role in prefrontal cortex function, it is not clear which 5-HT receptors exert these effects.

### 3.1. 5-HT1A Agonist Effects

The 5-HT1A agonism (ipsapirone) treatment alone was able to ameliorate the core symptoms of ADHD. This effect was mainly associated with enhanced expression of 5-HT2A and 5-HT1A neuroreceptors in all three regions investigated except for the dorsal striatum where 5-HT1A expression was reduced. It is well understood that 5-HT1A, being an inhibitory autoreceptor, decreases the release of 5-HT. Hence, it also enhances the 5-HT content in the synaptic cleft. For several decades, the 5-HT1A neuroreceptor has been utilized in pharmacological medication as an anxiolytic and antidepressant agent. In the present study, the ameliorating effect on anxiety-like behavior shown in the elevated plus maze and the open field test can be attributed to the anxiolytic effect exerted by 5-HTA agonism. Venlafaxine is a 5-HT and NE reuptake inhibitor that was able to ameliorate ADHD symptoms in adults [47]. A double blind randomized clinical trial with methylphenidate and buspirone (5-HT1A agonist) indicates efficacy of buspirone over MPH in hyperactive and impulsive domains of ADHD [48]. The 5-HT1A receptor gene can increase the susceptibility to ADHD [49]. Interestingly, 5-HT1A agonism also ameliorated locomotory hyperactivity and impulsivity in our study. 5-HT1A neuroreceptors are autoreceptors in the raphe nucleus while they act as postsynaptic neuroreceptors in other regions of the brain, including the prefrontal cortex. Hence, systemic drug effects may be attributed to either activation of pre- or postsynaptic 5-HT1A receptors or a combination of the two. The 5-HT1A and 5-HT2A neuroreceptors of the prefrontal cortex are also associated with inhibitory control and impulsivity [4]. The beneficial effect of 5-HT1A in reducing impulsivity may be due to activation of 5-HT1A in the prefrontal cortex. In our study, the expression of 5-HT1A was enhanced after 5-HT1A agonist treatment. In a recent study by Ochiai et al. [50], where they tested two 5-HT1A receptor agonists in mitigating hyperactivity in mice, 8-OH-DPAT was able to mitigate the hyperactivity. Wingen et al. [41] describe that the 5-HT2A receptor might be involved more in sustained attention compared to divided or selective attention. The DA-D1 receptor expression was reduced by the 5-HT1A agonist in all three regions while DA-D2 expression was reduced in the prefrontal cortex and the substantia nigra while increased in the dorsal striatum.

### 3.2. 5-HT2A Antagonism Effects

The 5-HT2A antagonist (MDL 100907) treatment in SHRs was able to ameliorate the core symptoms of ADHD. This beneficial effect was associated with enhanced expression of 5-HT2A receptors in the all three regions investigated. The 5-HT2A antagonist treatment reduced 5-HT1A receptors in the dorsal striatum and increased them in the substantia nigra while it did not have any effect on the prefrontal cortex. 5-HT2A facilitates the reuptake of serotonin from the synapse to the glial cells and postsynaptic membrane. Several studies have shown that reduction in 5-HT content in rat brains results in impulsive responses in 5-CSRT tasks [51,52] and this effect was enhanced by 5-HT2A antagonism [51]. Our findings are consistent with an earlier study in which decreased 5-HT2A receptors are associated with enhanced attention while their increased expression in the prefrontal cortex is associated with decreased inhibitory response [40]. Further, the role of 5-HT2A in the prefrontal cortex in impulsive response was demonstrated by many authors [42,43] where stereotaxic injection of 5-HT2A agonists into the medial prefrontal cortex resulted in increased impulsive responses. However, there are a few contradictory reports as well regarding the 5-HT2A antagonism. 5-HT2A blockade produces impairment of sustained attention and motor impulse control [41]. Blocking 5-HT2A receptors in the nucleus accumbens showed an enhanced premature response in 5-CSRT tasks [53]. As far as human studies are concerned, impulsivity in the form of premature responses was associated with high density of 5-HT2A neuroreceptors in the neocortical region measured with positron emission tomography (PET) [54]. This study on humans has also claimed that in ADHD the 5-HT2A receptor expression is unaffected, but 5-HT2A polymorphisms may be a modulating factor of the disease [55]. Not only is the 5-HT2A receptor functionally selective, but in vivo behavior can also vary between human and animal models. The most pronounced species differences are observed in rats, which are commonly used to model neuropsychiatric drugs. In rats, differences in the amino acid sequence of the fifth transmembrane domain 5 result in significant functional differences. In a recent study, the prefrontal cortex of ADHD rats expressed decreased 5-HT2A neuroreceptors while 5-HT1A neuroreceptors were increased compared to their WKY counterparts at baseline [56]. In our study, we did not find any such difference at the baseline. Further, they observed that glucocorticoid receptor treatment reduced hyperactivity in ADHD rats which is associated with enhanced 5-HT2A expression and reduced 5-HT1A expression in the prefrontal cortex, similar to our study results. Similar to this, Tanaka and co-workers [57] report that hyperactive behavior of mice was ameliorated by 5-HT1A antagonist WAY-100635. The 5-HT2A antagonist treatment reduced both DA-D1 and DA-DE receptors in the prefrontal cortex. Interestingly, it enhanced DA-D1 receptor expression in the substantia nigra and DA-D2 receptors in the striatum, while DA1 receptors were reduced in the striatum but enhanced in the substantia nigra.

### 3.3. Alpha-2 Agonist Effects

In the present study, the alpha-2 agonist (guanfacine) was able to ameliorate the core symptoms of ADHD like locomotor hyperactivity, anxiety-like behavior, impulsivity, and inattention. This effect was associated with elevated the expression of 5-HT1A and 5-HT2A receptors in the prefrontal cortex and the substantia nigra, while in the dorsal striatum, it reduced 5-HT2A expression. Our results are consistent with those of Friedman et al. [58] where treatment with the alpha-2A agonist (guanfacine) along with risperidone displayed enhancement of attention. Alpha-2A receptors in the brain are predominantly concentrated in the prefrontal cortex [59]. A moderate level of alpha-2A receptor stimulation improves prefrontal cortex regulation of attention [60]. In the present study, the alpha-2 adrenergic agonists reduced both DA-D1 and DA-D2 neuroreceptors in the prefrontal cortex and the substantia nigra while they increased in the dorsal striatum. Previously, alpha-2 adrenergic receptors caused inhibition of DA-D1 receptor signaling in striatonigral neurons and enhancement of DA-D2 receptor signaling in striatopallidal neurons [61]. The alpha-2 adrenergic receptor agonists indirectly modulate the DA receptor activities in both the prefrontal cortex and the substantia nigra. The possible cause of the opposite effects of DA receptor expression in the dorsal striatum and the substantia nigra needs to be further investigated. We postulated that the alpha-2 receptor agonism is necessary in therapeutic formulations to deal with ADHD as these receptors are vital in prefrontal-cortex-guided attention by inhibiting inappropriate behaviors.

### 3.4. Expression of DA Receptors

All three neuroreceptor manipulations downregulated both DA-D1 and DA-D2 receptors in the prefrontal cortex. However, the neuroreceptor manipulation had highly varying effects on the striatum and the substantia nigra. 5-HT1A agonist and alpha-2 adrenergic agonists downregulated both DA-D1 and DA-D2 receptors in the substantia nigra while 5-HT2A antagonist enhanced DA-D1 receptors. In the dorsal striatum, DA-D2 receptors were upregulated while DA-D1 receptors were downregulated except with alpha-2 adrenergic agonist treatment. The activation of the 5-HT1A and 5-HT2A receptors might have increased DA release in the medial prefrontal cortex, striatum, and substantia nigra. This might have resulted in subsequent inhibition of the glutamate and acetylcholine release in other areas of the brain [62]. Studies on animal models of ADHD indicate intimate interplay between 5-HT and dopaminergic neurotransmission [63]. The DA-D1 receptors are found mainly in the postsynaptic membrane where they regulate the postsynaptic transmission of DA. They are expressed in excitatory neurons in the prefrontal cortex [64]. It has been shown that the hypofunction of DA-D1-receptor-mediated regulation of GABAergic inhibitory synaptic transmission in the cingulate cortex might play a role in the pathophysiology of ADHD [65]. Bilateral microinjection of both DA-D1 and DA-D2 receptor antagonists into the prefrontal cortex of mice resulted in impairment of attention [66]. Blocking of DA-D1 and DA-D2 receptors is expected to enhance the DA level in the synaptic cleft and reduce inattention, however, the authors of [66] claim attention was impaired, and the possible cause was not addressed. Probably, blocking DA-D1 receptors at the presynaptic membrane would have enhanced or normalized DA levels in the synaptic cleft. In our study, we observed reduced DA receptor expression in the prefrontal cortex as a part of the neuroadaptation due to chronic receptor manipulation.

The DA-D2 dopamine receptors are highly expressed in the prefrontal cortex, responsible for locomotion, reward, reinforcement, memory, and learning [67]. The DA-D2 receptors are found in both presynaptic and postsynaptic membranes and facilitate the reuptake of DA from the synapse to the presynaptic membrane via DA-D1 receptors. DA secreted in the prefrontal cortex binds and activates DA-D2 receptors on the postsynaptic membrane [68]. The activated DA-D2 receptors phosphorylate Akt, an intracellular signaling protein that targets mTOR [69]. The latter activates S6 kinase, a transcription factor initiating transcription and translation of certain genes in the PFC, allowing the synthesis of certain proteins [70]. These proteins increase the connection between the prefrontal cortex and other parts of the brain and thus improve attention and reduce hyperactivity and impulsivity. However, very low or very high DA secretions in ADHD cases will prevent DA-D2 receptors from phosphorylating Akt. Overexpression of DA-D2 receptors in the striatum affects dopamine levels, rates of dopamine turnover, and activation of DA-D1 receptors in the prefrontal cortex [71], events that are critical for attention. Deletion of DA-D2 receptors in a mouse model of ADHD showed hyperactivity and increased reward behavior [72]. It has been shown that the DA-D2 receptor expression levels in the substantia nigra and striatum were increased in ADHD rats [73]. In our study, though we did not observe any difference at the baseline between SHRs and WKY rats, the neuroreceptor manipulation was able to reduce the DA-D2 receptors in the substantia nigra while ameliorating ADHD core symptoms. This indicates that ADHD may be associated with higher expression of DA-D2 receptors in the substantia nigra, so downregulating their expression would be beneficial. It has also been hypothesized that increasing the number of DA receptors to normal levels reduces impulsivity and anhedonia symptoms. This is due to the fact that reduced DA receptors in the substantia nigra are associated with impulsive behavior and also addictive behavior [74]. However, the study did not specify the subtype of dopamine receptors involved. Our study reveals that ameliorating ADHD behavior is predominantly associated with downregulation of DA-D1 and DA-D2 receptors in the areas of the brain concerned with ADHD and upregulation of DA-D2 receptors in the dorsal striatum. The relatively potent antagonism of serotonin 5-HT2A receptors coupled with relatively weaker antagonism of DA-D2 receptors is the central pharmacological characteristic shared by most of second-generation antipsychotics. The results of the present study indicate that both a 5-HT2A antagonist and 5-HT1A agonist are good enough in ameliorating ADHD symptoms by indirectly acting on DA-D1 and DA-D2 receptors in spontaneously hypertensive rats.

### 3.5. Involvement of the Striatum, Prefrontal Cortex, and the Substantia Nigra in ADHD

The striatum and the prefrontal cortex are the major regions of the brain involved in the two major neuronal networks (frontostriatal and mesocortical) associated with ADHD. The frontostriatal network connects the prefrontal cortex with the striatum which is involved in inattention, hyperactivity, impulsivity, and cognitive functions [75]. The mesocortical pathway is dopaminergic and consists of neurons projecting from the anterior tegmental area of the midbrain to the prefrontal cortex (the mesolimbic pathway connects with striatum, nucleus accumbens, olfactory tubercle, and amygdala). The connection with the prefrontal cortex is concerned with attention, on-task behavior, and on-task cognition. Inadequate DA and NE contents in both the prefrontal cortex and the striatum are characteristics of ADHD. Reduced DA receptors in presynaptic (axonal terminals ending in the prefrontal cortex) or enhanced postsynaptic membranes (dendrites of prefrontal cortex neurons) are likely to reduce the extracellular DA level which is the primary cause of ADHD. Activated presynaptic DA receptors and inhibited postsynaptic DA receptors enhance the extraneuronal DA content and minimize the ADHD core symptoms like inattention, hyperactivity, and impulsivity. The specific neuronal network and the neurotransmitters utilized in regulating the different features of cognition like attention, inhibition of inappropriate behaviors, and emotional regulation through the prefrontal cortex have been well established [76]. NE stimulation of alpha-2A receptors enhances prefrontal cortex function by strengthening appropriate network connections and DA stimulation of DA-D1 receptors exerts its beneficial effects by weakening inappropriate connections [77]. In addition to DA and NE, a lower level of 5-HT in the prefrontal cortex is also associated with ADHD. The prefrontal cortex contains a very high density of 5-HT1A (inhibitory) and 5-HT2A (excitatory) receptors [77]. In the present study, the 5-HT1A agonist and alpha-2 adrenergic agonist enhanced 5-HT1A and 5-HT2A receptor expression while DA-D1 and DA-D2 expression was reduced, probably indicating higher utilization of serotonin in the prefrontal cortex. The DA receptors of the striatum and the substantia nigra were mainly concerned with motor activities. The expressions of neuroreceptors were highly varied in the striatum. The DA-D1 receptors, which are mainly excitatory to the striatal neurons, and their expression were reduced in the dorsal striatum but also in the prefrontal cortex and the substantia nigra. The net effect of the excitatory striatal neurons will be inhibitory to the cortex. The DA-D2 receptors, which are mainly inhibitory to the striatal neurons, and their expression were enhanced. In the present study, the DA-D2 receptors were reduced in the dorsal striatum. These inhibited neurons of the striatum will send excitatory fibers to the cortex via the thalamus. Hence, reduced DA-D2 receptors in the striatum might have reduced the excitatory fibers to the cortex in controlling hyperactivity in SHRs.

Considering the heterogeneity and multifactorial etiology of ADHD, the treatment of ADHD remains highly challenging for pediatric psychiatrists. Stimulant medication and a variety of second-generation antipsychotics are used in children with ADHD based on the severity of ADHD symptoms. The second-generation antipsychotics often utilize DA receptor antagonism. In our study, we have demonstrated that 5-HT1A agonism and 5-HT2A antagonism can act as monotherapies in tackling the core symptoms of ADHD by indirectly modulating DA neuroreceptor activity. One of the limitations of our study is that we did not measure the extracellular DA, NE, and 5-HT levels in the areas of the brain affected in ADHD. This could have added more weight to the results observed in this study. No studies have demonstrated the effect of atypical antipsychotics on the expression of various neuroreceptors in the regions of the brain affected in ADHD except for a human study focusing on atomoxetine. It is important to note that findings in animal models do not always directly translate to humans, and the complex interplay between neurotransmitter systems makes it challenging to draw straightforward conclusions. Nevertheless, this work only confirmed the correlations among 5-HT1A, 5-HT2A, and D1/D2 in ADHD at this stage, and it is certainly important to reveal the mechanism of ADHD pathogenesis at a molecular level and to help the discovery of new drug targets for the treatment of ADHD in the future. We conclude that the 5-HT1A agonist and 5-HT2A antagonist monotherapies alone could reduce the core symptoms of ADHD by altering the expression of DA1 and DA2 neuroreceptors in prefrontal cortex, striatum, and substantia nigra.

## 4. Materials and Methods

### 4.1. Animals

Fifteen-day-old male spontaneously hypertensive rats (SHRs) and Wistar-Kyoto (WKY) rat pups, bred and maintained at the Animal Resources Center at the Health Sciences Center (HSC), Kuwait University, were used in this study. Spontaneously hypertensive rats were originally obtained from Charles River Laboratories (Wilmington, MA, USA) and maintained in the Animal Resources Center, HSC, Kuwait University. These rats were housed in sterile polypropylene cages in a controlled environment (22 ± 2 °C) with a 12 h light/dark cycle. The rats were kept with the mother till they attained the age of 21 days and then they were weaned and fed on standard food and water *ad libitum*. The animal treatment protocol and maintenance were according to the approved protocol of the Institutional Animal Care and Use Committee of Kuwait University, which follows the recommendations of NIH Guidelines and the Guide for the Care and Use of Laboratory Animals (approval letter No. 23/VDR/EC/3560, 7 November 2019). All efforts were made to minimize the number of animals used and their suffering.

### 4.2. Experimental Design

Fifteen-day-old SHR pups and Wistar-Kyoto (WKY) rat pups were used in the experiment. Fifteen-day-old SHRs and Wistar-Kyoto (WKY) rat pups were divided into eight groups: (i) normal control (SHR-NC), (ii) Wistar-Kyoto normal control (WKY-NC), (iii) SHR-MDL (MDL 100907), (iv) WKY-MDL, (v) SHR-IPS (ipsapirone), (vi) WKY-IPS, (vii) guanfacine hydrochloride (SHR-GFC), and (viii) WKY-GFC (n = 12 in each group). Rat pups in each group were treated with receptor agonists or antagonists (see Section 4.3 for details) for four weeks (from postnatal day 15 (PND 15) to PND 42). As the treatment started on PND 15, the rat pups continued to be with their mother till weaning on postnatal day 21. Thereafter, they were housed in separate cages (2–3 rats/cage). Age-matched untreated SHR pups (SHR-NC group) served as ADHD models. Age-matched normal *Wistar-Kyoto* (WKY) rat pups served as a control group (WKY-NC group). The age of the animals used in this study (PND 15–43) for the treatment was equivalent to childhood and early adolescence in humans [78].

Rat pups in all groups were subjected to open field, locomotor activity, impulsivity in a modified T-maze, impulsivity for water drinking in aversive electro foot shock apparatus and elevated plus maze behavioral tests over a period of three weeks after the last day of drug administration (PND 44–65). At the end of all behavioral tests (on PND 65), rats in all groups were either perfused with 4% paraformaldehyde for immunohistochemical studies on expression of 5-HT1A, 5-HT1A, DA-D1, and DA-D2 receptors in the prefrontal cortex, striatum, and substantia nigra (n = 6/group) or fresh brain tissue was collected for estimation of 5-HT1A, 5-HT1A, DA-D1, and DA-D2 receptor quantity in the prefrontal cortex, striatum, and substantia nigra tissue (n = 6/group).

### 4.3. Agonist and Antagonist Dose and Preparation

5-HT2A antagonist MDL 100907 (MDL, Cat. No. 4173), 5-HT1A agonist ipsapirone (IPS, Cat. No. 1869), and alpha-2A agonist guanfacine hydrochloride (GFC, Cat. No. 1030) were obtained from Tocris Bioscience, United Kingdom. Stock solution of each drug was prepared in DMSO and kept at −20 °C until use (concentration of stock solutions: MDL: 2 mg/mL, IPS: 4 mg/mL, GFC: 50 µg/mL). Working drug injection solutions were prepared daily just before use by diluting the stock solution in saline (concentration of working injection solutions: MDL: 100 µg/mL, IPS: 200 µg/mL). Drugs were administered intraperitoneally. MDL, IPS, and GFC were given at doses of 0.5 mg/kg, 1 mg/kg, and 0.3 mg/kg, respectively, during the first week, 0.75 mg/kg, 1.5 mg/kg, and 0.45 mg/kg, respectively, during the second week, and 1 mg/kg, 2 mg/kg, and 0.6 mg/kg, respectively, during the third and fourth weeks. The dose selected was based on previous studies [79,80]. The amount of drug injected for each rat was calculated based on the body weight at the beginning of each week. For control animals, an equivalent volume of saline with DMSO (50 µL DMSO/mL saline) was injected.

### 4.4. Behavioral Tests (for Assessment of ADHD)

All behavioral tests were performed on all rats in all groups (from PND 44–PND 65). Open field test apparatus and locomotor activity test apparatus were used for measuring locomotor activity. Open field test apparatus and elevated plus maze apparatus were used for assessment of anxiety level. A modified T-maze and aversive electro foot shock apparatus with water reward were used for testing impulsive behavior.

#### 4.4.1. General Locomotor and Exploratory Behavior in a Novel Environment

To evaluate the general locomotor and exploratory activities, an open field test was used [81]. The open field test was to test exploratory locomotor activity and anxiety of the animals. The open field test apparatus was a box made of Plexiglas having dimensions of 90 cm (length) × 90 cm (breadth) × 20 cm (height). Apparatus was kept under the camera of an EzVideoTM video tracking system (V5.70, Accuscan Instruments, Columbus, OH, USA). Floor area was virtually divided into central and peripheral zones and calibrated. A rat was placed in the center of the apparatus and allowed to move freely for 10 min, and movement was tracked by the video tracking system. Distance traveled and time spent in the central and peripheral zones recorded by the video tracking system were analyzed using the video tracking software. Increased total distance traveled in the given time is an indicator of hyperactivity and increased distance traveled in the peripheral zone of the open field test apparatus is suggestive of enhanced anxiety level [81].

#### 4.4.2. Locomotor Hyperactivity Assessment

To measure locomotor hyperactivity for a long period of time, in a familiar environment, all rats were tested in a locomotor activity test apparatus for 60 min. The apparatus was a black Plexiglas box having dimensions of 60 cm (L) × 60 cm (B) × 60 cm (H). The apparatus was kept under the video tracking system. To guarantee the novelty of the cage did not influence the activity measurements, the first 15 min of activity were not considered [81]. Animal movements were recorded for 60 min. The movement time (amount of time when the rat was moving), number of movement events (a movement event is defined as one or more changes in position greater than a certain distance, within three seconds of one another), rest time (amount of time when the rat was stationary), and distance traveled data recorded by the video tracking system were analyzed using video tracking software.

#### 4.4.3. Anxiety-like Behavior Assessment in Elevated plus Maze

An elevated plus maze test was used to evaluate anxiety-like behavior in rats in all groups. Elevated plus maze apparatus consisted of a plus maze with four arms extending from a central intersecting area. Each arm was 30 cm long and 10 cm wide and supported by a leg of 30 cm in height. Two opposite arms were without any side walls (open arm) and the other two arms had a side wall of 30 cm high. The plus maze was placed on a table, under the video tracking camera. Rats were placed in the intersection of the four arms of the elevated plus maze and allowed to explore the arms for eight minutes. Time spent in open and enclosed arms was recorded for eight minutes using the video tracking system. The percentage of time spent in each open and closed arm was calculated using the formula given below [82].
$$\%\;of\;time\;in\;the\;open\;arms=\frac{Time\;spent\;in\;the\;open\;arm\;(minutes)}{Time\;spent\;in\;the\;open\;arm+Time\;spent\;in\;the\;enclosed\;arm}\times 100$$

#### 4.4.4. Impulsivity Analysis in Modified T-Maze

Impulsivity was tested in a modified T-maze. The apparatus consisted of a T-maze with a start arm (30 cm (L) × 10 cm (W) × 30 cm (H) and two goal arms of the same dimensions as that of the start arm. There was a sliding door at the beginning of each arm. At the end of each goal arm (goal area), a food well was placed, which was separated from the stem by a sliding door [83,84]. Rats were deprived of food for 30–36 h before the commencement of the experiment, thereafter food was restricted to 5–6 (4–5 g) pellets/day which was given after the session of the day, such that body weight was maintained at 80% of the body weight before food deprivation. The impulsivity test consisted of three phases: (i) habituation phase, (ii) training phase, and (iii) testing phase. The habituation phase consisted of two sessions on the 1st day with two hours of intersession intervals (five trials/session). In each trial, three pellets were available in the goal area of each arm. A rat was released in the stem arm and allowed to explore both goal arms and eat the pellets in the goal area. Once the rat ate the pellets in the goal area of both goal arms or after a maximum of three minutes, the rat was returned to the start arm for the next trial. The training phase (2nd to 4th day) consisted of three sessions/day (five trials/session) with two hours of intersession intervals. During the training phase, in the right goal were three pellets (large reward) and in the left goal area one pellet (small reward) was available in each trial. A trial was started by placing the rat in the start arm and it was allowed to explore the arms. The rat was free to choose a large or small reward arm and eat the pellets available there. A trial ended when the rat ate the food in the arm it chose (i.e., it was not allowed to explore the other arm in that trial). Trials were repeated by placing the rat in the start arm after a one-minute intertrial interval. In each trial, arm choice was documented. The percentage of entries of each arm in each session was calculated (% entries into large reward arm = number of entries to large reward arm/5 × 100; % entries into small reward arm = number of entries to small reward arm/5 × 100). Sessions were continued on three successive days till rats chose the large reward arm at least 80% of the time in two consecutive sessions (i.e., in a session of five trials, the rat must select the large reward arm four times, in two consecutive sessions). This took about 7–9 sessions. The test phase started only after the rat had reached the above criteria in the training phase and continued for five more sessions over the next two days. The number of pellets in each arm was as in the training phase (three pellets and one pellet). A trial was started by placing the rat in the start arm and it was allowed to enter the small or large reward arms. If the rat entered the small reward arm, it was free to go to the goal area and eat the pellet there. The trial ended with that. If the rat entered the large reward arm, it was detained in that arm (by closing the sliding doors before the goal area and at the beginning of the arm) for 30 s before it was allowed to enter the goal area and eat the food there. The trial ended when the rat ate the pellets there. In each trial, the arm chosen by the rat was documented. The percentage of entries of each arm in each session was calculated (% entries into large but delayed reward arm = number of entries into large but delayed reward arm/5 × 100; % entries into small reward arm = number of entries into small reward arm/5 × 100). A decreased % of entries into the large but delayed reward arm is suggestive of impulsive behavior [83,84].

#### 4.4.5. Impulsivity Analysis in Electro Foot Shock Aversive Water Drinking Test (EFSDT)

Impulsivity was tested in the electro foot shock aversive water drinking test (EFSDT) box [81]. The EFSDT box measured 40 × 40 × 40 cm. The box was made of Plexiglas, painted black. The floor area was made up of an electrified metal grid, and it was connected to an electrical foot shock source. The floor area was divided into four regions: start box (10 cm × 10 cm), choice area (40 cm × 20 cm), water area (20 cm × 20 cm), and dummy (no) water area (20 cm × 20 cm). Water and dummy water areas were separated from each other by a Perspex partition. The start box area was connected with the choice area, which could be closed by a sliding door. The choice area was connected with both the water area and dummy water area. In the water area and dummy water area, a water bottle with a stainless-steel nozzle was fitted from outside of the box so that the nozzle extended 4 cm into the box at a height of 6 cm above the floor. During the experiment, the water bottle on the water area side was filled with water but not the dummy water area side. The apparatus was kept under the EzVideoTM video tracking system for video tracking of the rats.

Rats were deprived of water for 30–36 h before the commencement of the experiment, thereafter water was given for 15 min after the session of the day. The experimental procedure consisted of three phases: (i) habituation phase, (ii) training phase, and (iii) testing phase. The habituation phase consisted of one session (five trials/session) on the 1st day. In each trial, a rat was released in the start box, the door was opened, and the rat was allowed to explore both water and dummy water areas for 3 min. Water was not available in neither the water nor dummy water area. The training phase consisted of four sessions of 5 trials each (two sessions/day, with two hours of intersession interval, intertrial interval = 1 min). In each trial, a rat was released in the start box, the door was opened, and it was free to explore both the water and dummy water areas. The trial ended once the rat entered the water area and drank water there for 5 s or entered the dummy water area and stayed there for more than 3 s. The number of entries into the water area and hence % of water area entry frequency were calculated for each session. The testing phase started after the rats entered the water area 80% of the time (4 entries into the water area in 5 trials of the session). Normally, all rats achieved this in 3–4 sessions. Only after the rat had reached the above criteria in the training phase did the impulsivity testing phase start. The testing phase consisted of 2 sessions of 5 trials each. The protocol during the impulsivity testing phase was the same as in the training phase, however, each time the rat attempted to drink water, a foot shock was given (2 mA, for 0.5 s) and the trial was ended. The number of entries into the water area was collected from the video tracking data and hence % of water area entry frequency was calculated for each session. The number of impulsive water drinking attempts was documented manually. Increased % of entry into the water area and increased number of water drinking attempts are suggestive of impulsive behavior [81].

### 4.5. Western Blotting

For Western blot analysis, the rats were euthanized with CO_2_, perfused with 100 mL of ice-cold saline, and the brain was dissected rapidly on a cold dissection table. The prefrontal cortex, striatum (dorsal striatum), and substantia nigra tissue were dissected in cold saline, weighed, and snapped frozen in liquid nitrogen. Tissue samples were stored at −80 °C until analysis. Tissue samples were thawed to 4 °C and homogenized in a known volume of radioimmunoprecipitation assay (RIPA) buffer containing 50 mM Tris, 1%NP-40, 5 mM EDTA 0.1%SDS, protease inhibitors (100 µL/mL), 2 mM benzamidine, PMSF (10 µL/mL), and 0.5% Na-deoxycholate (Sigma Chemicals). Homogenate was centrifuged at 14,000× *g* for 20 min in a refrigerated (4 °C) centrifuge. Protein level in the sample was measured by the Bradford method. Samples were then denatured by boiling for 5 min in a boiling water bath. The soluble protein fractions in the sample were separated by electrophoresis in a 10% SDS-polyacrylamide gel (30 µg protein/well) and electro blotted onto a PVDF membrane (Millipore, Bedford, MA, USA) in a blotting apparatus (Bio-Rad Laboratories, Inc., Hercules, CA, USA). The membrane was blocked with 5% dried skim milk, 0.05% Tween-20 in tris-buffered saline, and Tween-20 (TBST) at room temperature for 2 h and incubated with appropriately diluted primary antibodies (5-HT1A receptor (rabbit polyclonal anti-5HT1A receptor-abcam-85615); 5-HT2A receptor (rabbit polyclonal anti-5HT2A receptor-abcam-66049); DA receptor-1 (DA1, rabbit polyclonal anti-DA-D1 receptor-ab20066), DA receptor-2 (DA-D2, rabbit polyclonal anti-DA-D2 receptor- ab21218) overnight at 4 °C on a shaker. The membrane was washed three times with TBST, and then incubated with appropriately diluted HRP-conjugated secondary antibodies (goat anti-rabbit IgG-HRP-ab205718) for 2 h at room temperature and washed with TBST. The PVDF membrane was treated with enhanced chemiluminescent substrate (ECL) and exposed to an X-ray hyper-film (5-HT1A Western blotting [85]; 5-HT2A Western blotting [86]; DA-D1 and DA-D2 Western blotting [87]). To confirm equal protein loading, the same blots were reincubated with antibodies specific for β-actin (Abcam, Cambridge, UK; 1:1000). Immunoreaction for β-actin was detected with ECL. The densities of the protein bands were measured in a GS-800 calibrated densitometer and normalized to respective β-actin values [88].

### 4.6. Immunohistochemistry

For immunohistochemistry, rats were euthanized with CO_2_ and perfused transcardially with 100 mL of phosphate-buffered saline (PBS), followed by freshly prepared 4% paraformaldehyde solution in 100 mM phosphate buffer (PB, pH 7.4). The brains were dissected and postfixed in the same fixative for 48 h. Brains were cryoprotected in graded sucrose solutions (10, 20, and 30%, 24 h in each) Serial coronal sections, 30 μm in thickness, were cut with a cryostat (Leica, Nußloch, Germany) and collected in 24-well culture plates filled with phosphate buffer. Free-floating sections corresponding to the regions of the prefrontal cortex, striatum (caudate–putamen), and substantia nigra of the midbrain were selected and processed for immunostaining for 5-HT1A, 5-HT2A, DA-D1, and DA-D2 receptors. The sections were treated with 3% H_2_O_2_ for 30 min to reduce the endogenous peroxidase activity in the tissue. Sections were incubated for 30 min with appropriate blocking serum along with 0.3% Triton X-100 to block the non-specific binding of the primary antibody to the tissue. The sections were then incubated with appropriately diluted primary antibodies (the same antibodies used for Western blot) overnight at 4 °C. The sections were incubated with the appropriately diluted biotinylated goat anti-rabbit secondary antibody for 1 h. The sections were treated with avidin–biotin–peroxidase complex (ABC) for 1hr and subsequently color was developed with 3,3′-diaminobenzidine (DAB). Throughout the protocol, sections were washed three times with PBS after each incubation. To assess non-specific staining, several sections in each experiment were incubated in a buffer without primary antibody. After immunostaining, sections were mounted on gelatinized slides and dried overnight. Sections were lightly counter stained with hematoxylin, dehydrated in ethanol grades, cleared in xylene, and cover slipped in Permount (Fisher Scientific, Pittsburgh, PA, USA). High-quality images were captured with a 40× objective with an Olympus digital camera (DP75) attached to an Olympus microscope. In the prefrontal cortex region, medial, lateral, and orbital regions were evaluated. In the striatum, the dorsal part was considered for immunostaining (5-HT1A receptor immunochemistry [89]; 5-HT2A receptor immunochemistry [90]; DA-D1 and DA-D2 receptor immunochemistry [91]).

### 4.7. Statistical Analysis

The data were expressed as mean ± SD and analyzed with SPSS (version 25) statistical analysis software. Data were analyzed with two-way ANOVA, followed by Bonferroni’s multiple comparison post hoc test. *p* values < 0.05 were considered as significant.

## Figures and Tables

**Figure 1 ijms-25-02300-f001:**
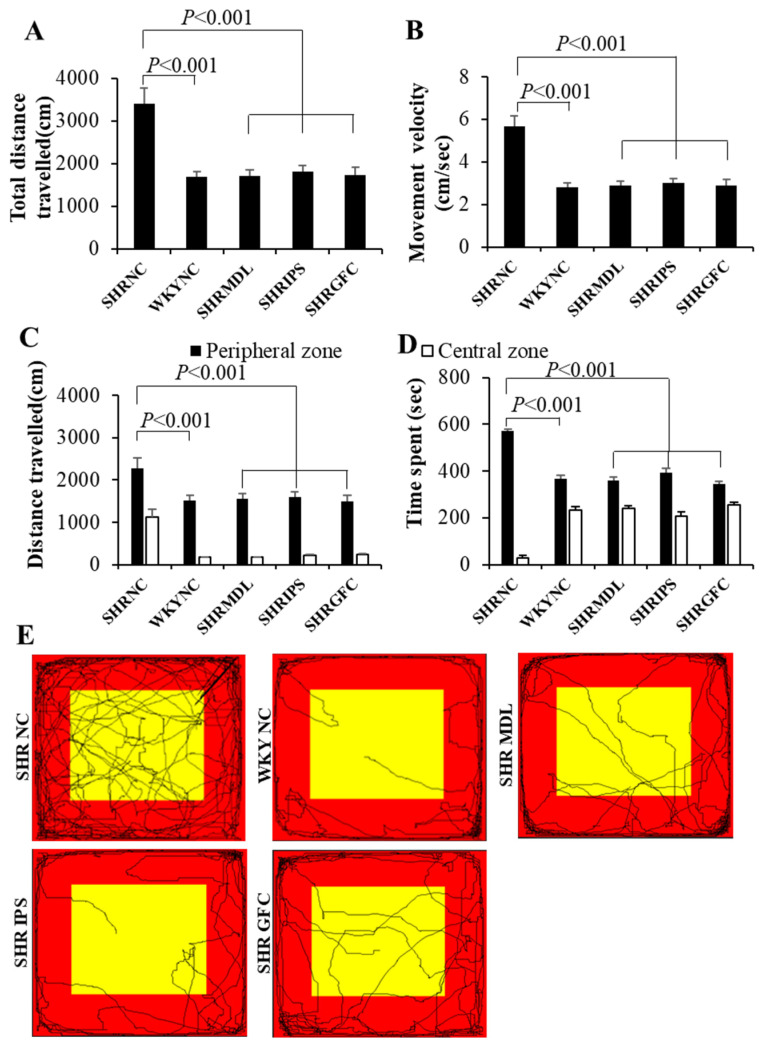
(**A**,**B**): Total distance traveled (**A**), movement velocity (**B**) of rats in different groups in open field test. Note significantly increased total distance traveled and movement velocity in SHR-NC compared to WKY-NC rats and they were significantly decreased in SHR-MDL, SHR-IPS, and SHR-GFC groups compared to SHR-NC group. (**C**,**D**): Distance traveled (**C**) and time spent (**B**) in the peripheral and central zones by rats in different groups in open field test. Note significantly increased distance traveled and time spent in the peripheral zone by SHR-NC compared to WKY-NC rats and they were significantly decreased in SHR-MDL, SHR-IPS, and SHR-GFC groups compared to SHR-NC group. (**E**): Representative video tracking of rats in different groups in open field test. Note significantly more movements in SHR-NC compared to WKY-NC rats. Movements in central and peripheral zones decreased in SHR-MDL, SHR-IPS, and SHR-GFC groups compared to SHR-NC group. (Red: peripheral zone, yellow: central zone).

**Figure 2 ijms-25-02300-f002:**
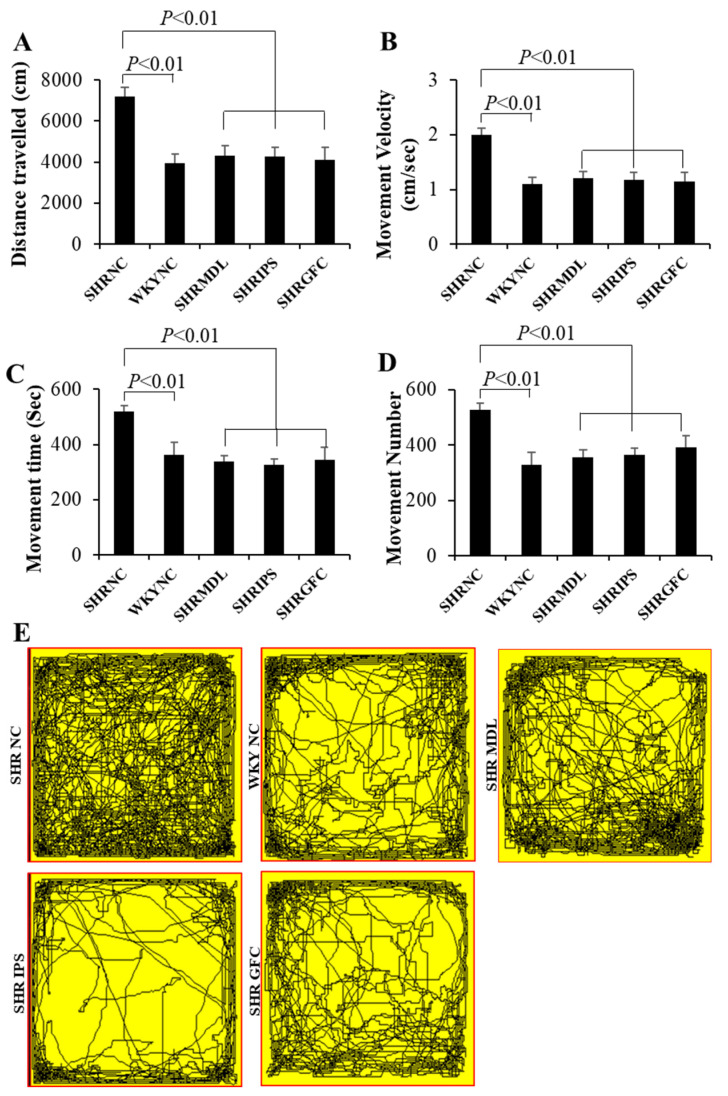
(**A**,**B**) Distance traveled (**A**) and movement velocity (**B**) of rats in different groups in locomotor activity test. Note significantly increased distance traveled and movement velocity of SHR-NC compared to WKY-NC rats, and they were significantly decreased in SHR-MDL, SHR-IPS, and SHR-GFC groups compared to SHR-NC group. (**C**,**D**): Movement time and movement numbers (**D**) of rats in different groups in locomotor activity test. Note significantly increased movement time and movement numbers in SHR-NC compared to WKY-NC rats and they were significantly decreased in SHR-MDL, SHR-IPS, and SHR-GFC groups compared to SHR-NC group. (**E**): Representative video tracking of rats in different groups in locomotor activity test. Note significantly increased movement in SHR-NC compared to WKY-NC rats and it was significantly decreased in SHR-MDL, SHR-IPS, and SHR-GFC groups compared to SHR-NC group.

**Figure 3 ijms-25-02300-f003:**
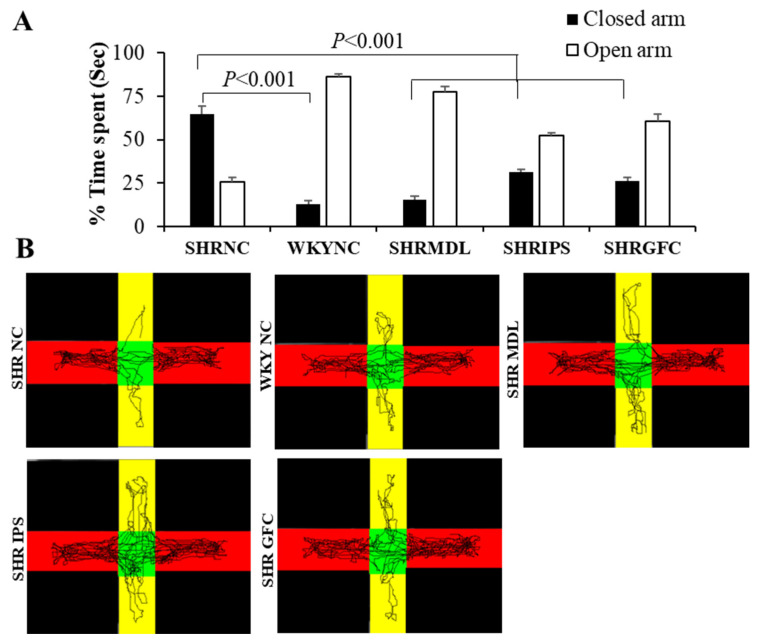
(**A**). % time spent in the open and closed arms of the elevated plus maze apparatus during elevated plus maze test by rats in different groups. Note significantly increased % time spent by SHR-NC in closed arm compared to WKY-NC rats and it was significantly decreased in SHR-MDL, SHR-IPS, and SHR-GFC groups compared to SHR-NC group. (**B**). Representative video tracking of rats in different groups during elevated plus maze test. Note significantly increased exploration by SHR-NC in closed arm (red) compared to WKY-NC rats and it was significantly decreased in SHR-MDL, SHR-IPS, and SHR-GFC groups compared to SHR-NC group (yellow: open arm).

**Figure 4 ijms-25-02300-f004:**
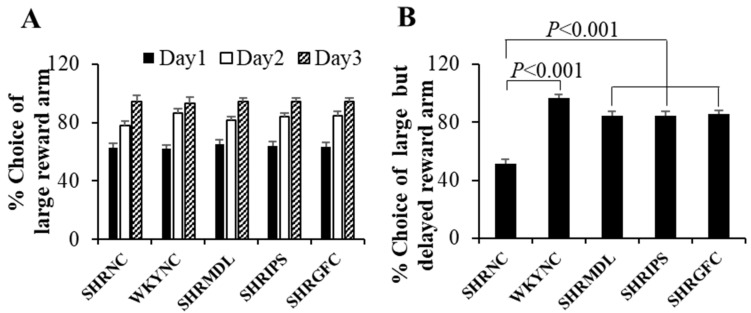
(**A**). Mean % choices of large reward arm during learning sessions on days 1, 2, and 3 and (**B**). mean % choice of large but delayed reward arm during test session by rats in different groups in modified T-maze test for impulsivity. There was a progressive increase in the choice of large, rewarded arm from day 1 to day 3 during learning phase in all groups. Note significantly fewer choices of large but delayed reward arm by SHR-NC rats in test session compared to WKY-NC rats and significantly increased choice of large but delayed reward arm by SHR-MDL, SHR-IPS, and SHR-GFC groups on test day compared to SHR-NC group.

**Figure 5 ijms-25-02300-f005:**
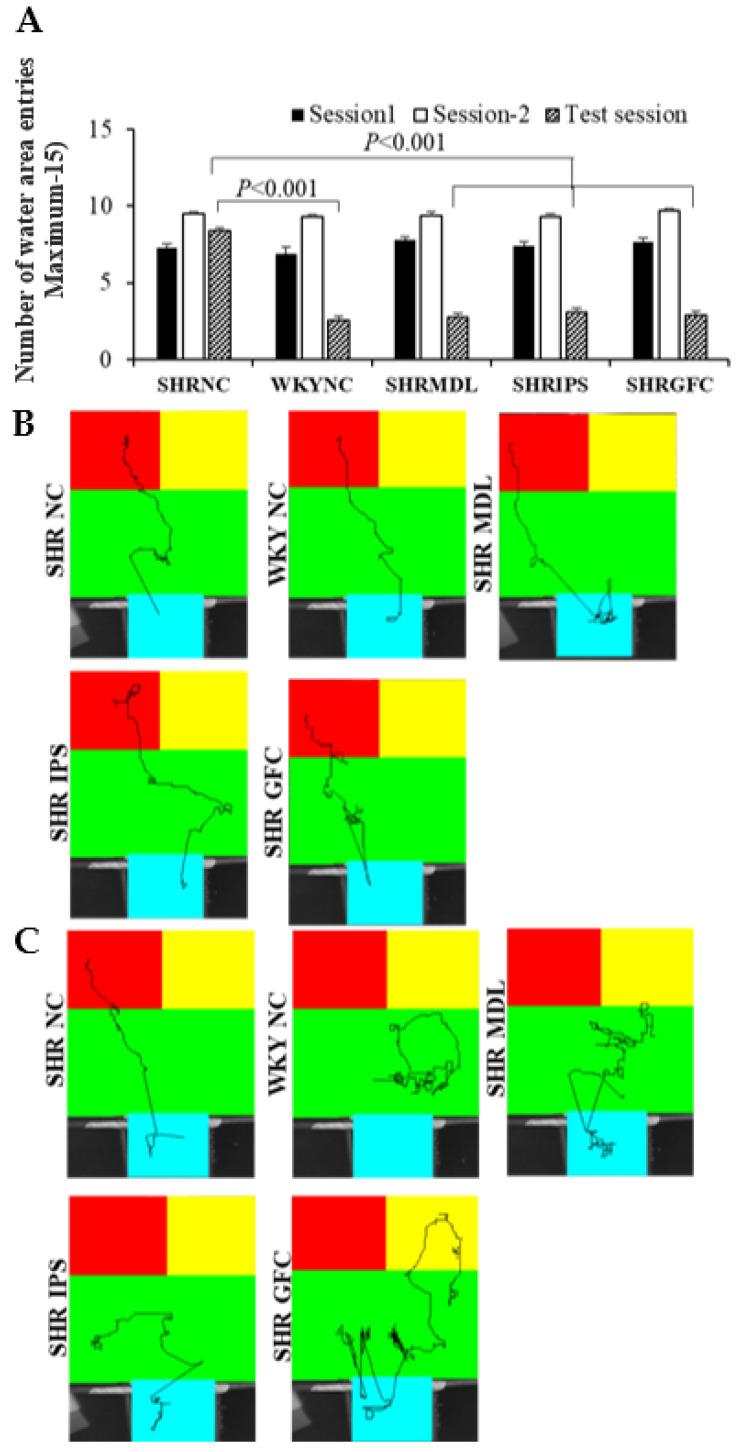
(**A**). Water area entry frequency during training sessions 1 and 2 and during impulsive drinking test session in impulsivity test in aversive electro foot shock apparatus. During 2nd training sessions, rats in all groups entered the water area in more than 90% of trials. Note significantly higher frequency of entry into water area (and impulsive water drinking) in spite of foot shock for each drinking act by SHR-NC rats in test session compared to WKY-NC rats. However, no such impulsivity was found in SHR-MDL, SHR-IPS, and SHR-GFC groups on test day compared to SHR-NC group. (**B**). Representative video tracking of rats in different groups in impulsivity test session in aversive electro foot shock apparatus during last training session, and (**C**). test session. During 2nd training sessions, rats in all groups entered the water area in more than 90% of trials. (Blue: start area, green: choice area, red: water area, yellow: no water area).

**Figure 6 ijms-25-02300-f006:**
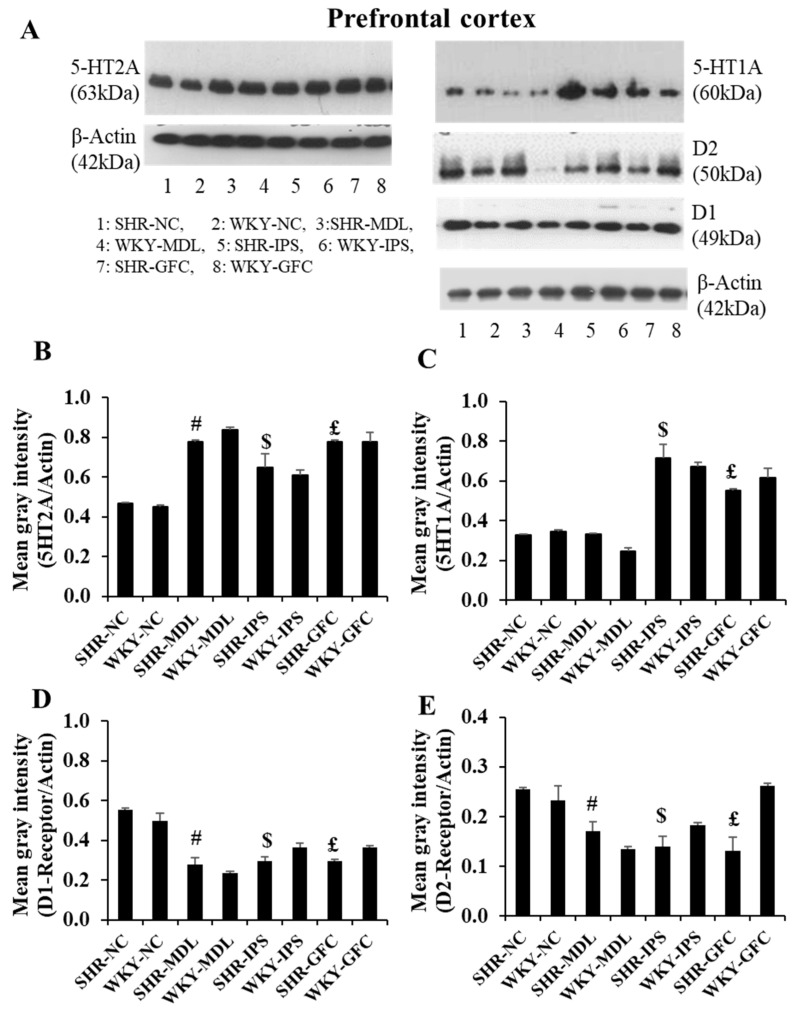
(**A**): Immunoblots of 5-HT2A, 5-HT1A, DA-D1, and DA-D2 receptors in the prefrontal cortical tissues. (**B**–**E**): Mean gray intensity of 5-HT2A (**B**) and 5-HT1A (**C**), DA-D1 (**D**), and DA-D2 (**E**) receptor immunobands normalized to actin band gray intensity. # SHR-NC vs. SHR-MDL, *p* < 0.05; $ SHR-NC vs. SHR-IPS, *p* < 0.05; £ SHR-NC vs. SHR-GFC, *p* < 0.05.

**Figure 7 ijms-25-02300-f007:**
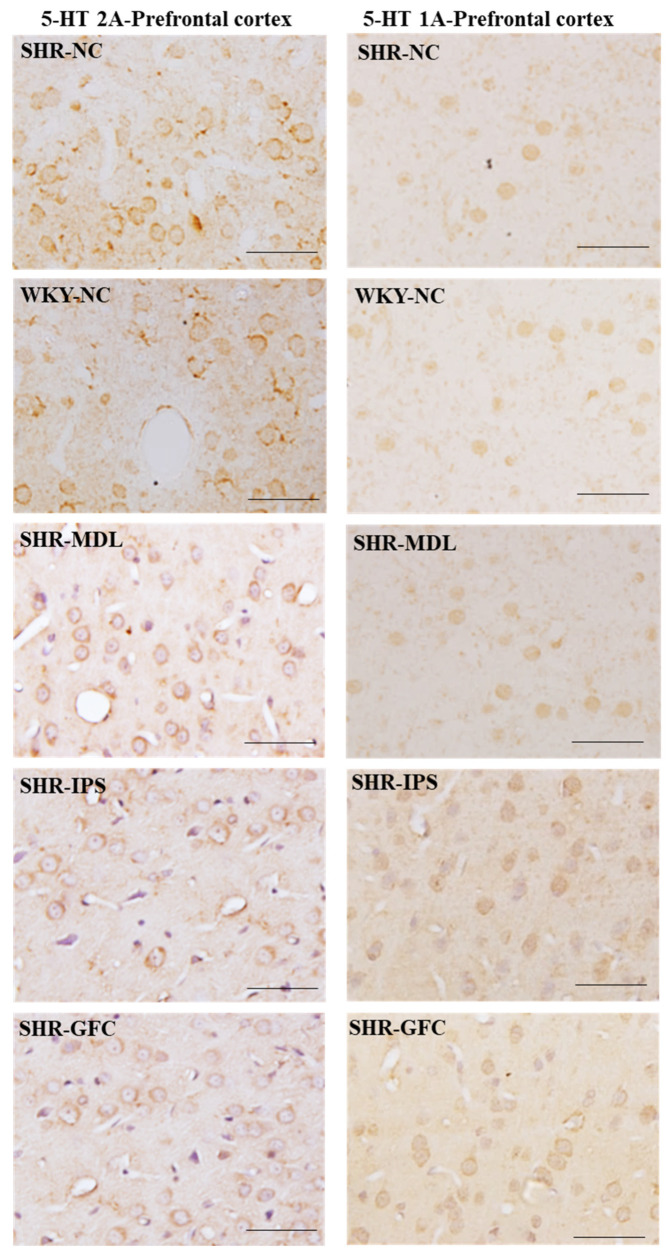
Photomicrographs of prefrontal cortical sections immunostained for 5-HT2A (**left** panel) and 5-HT1A (**right** panel) receptors. Note the expression pattern of 5-HT2A and 5-HT1A receptors in different groups which agrees with mean gray intensity of immunoblot of Western blot analysis shown in Figure 6B,C. Scale bar = 50 µm.

**Figure 8 ijms-25-02300-f008:**
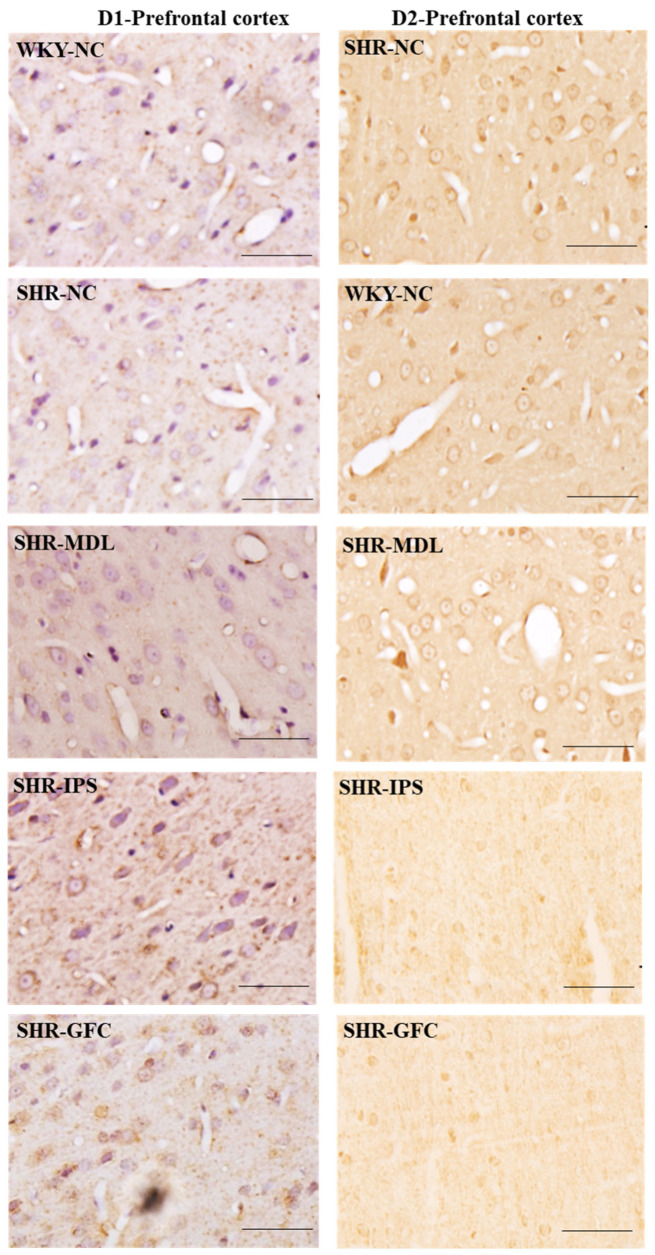
Photomicrographs of prefrontal cortical sections immunostained for DA-D1 (**left** panel) and DA-D2 (**right** panel) receptors. Note the expression pattern DA-D1 and DA-D2 receptors in different groups which agrees with mean gray intensity of immunoblot of Western blot analysis shown in Figure 6C,D. Scale bar = 50 µm.

**Figure 9 ijms-25-02300-f009:**
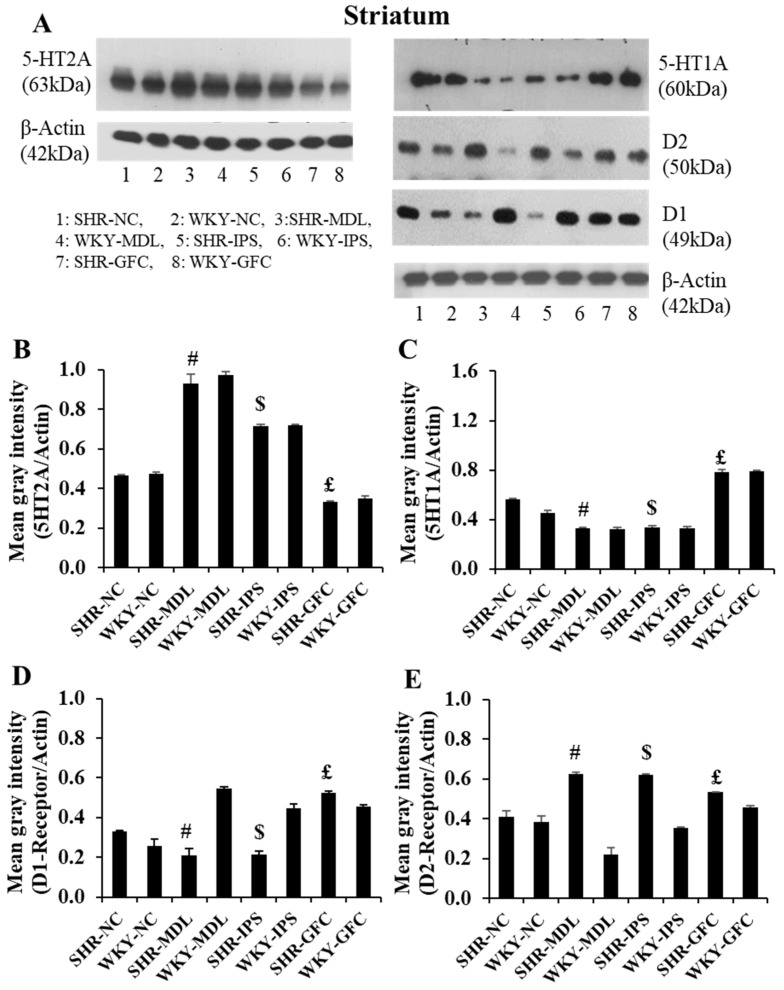
(**A**): Immunoblots of 5-HT2A, 5-HT1A, DA-D1, and DA-D2 receptors in the striatal tissues. (**B**–**E**): Mean gray intensity of 5-HT2A (**B**) and 5-HT1A (**C**), DA-D1 (**D**), and DA-D2 (**E**) receptor immunobands normalized to actin band gray intensity. # SHR-NC vs. SHR-MDL, *p* < 0.05; $ SHR-NC vs. SHR-IPS, *p* < 0.05; £ SHR-NC vs. SHR-GFC, *p* < 0.05.

**Figure 10 ijms-25-02300-f010:**
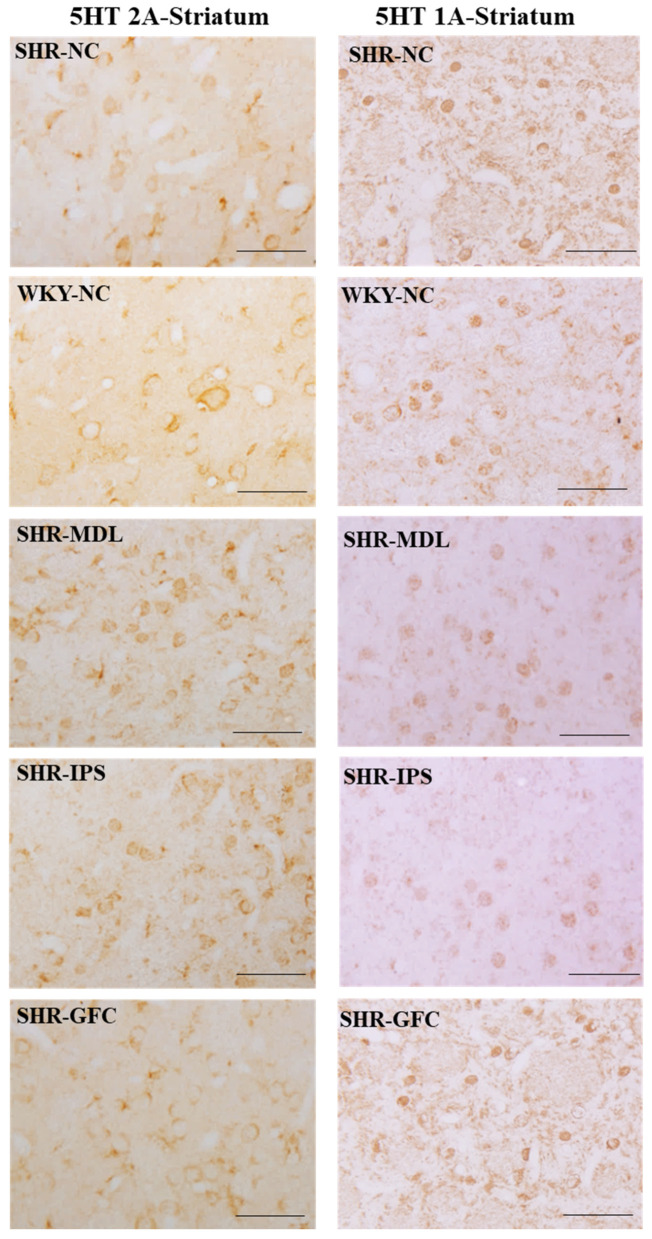
Photomicrographs of prefrontal striatal sections immunostained for 5-HT2A (**left** panel) and 5-HT1A (**right** panel) receptors. Note the expression pattern 5-HT2A and 5-HT1A receptors in different groups which agrees with mean gray intensity of immunoblot of Western blot analysis shown in Figure 9B,C. Scale bar = 50 µm.

**Figure 11 ijms-25-02300-f011:**
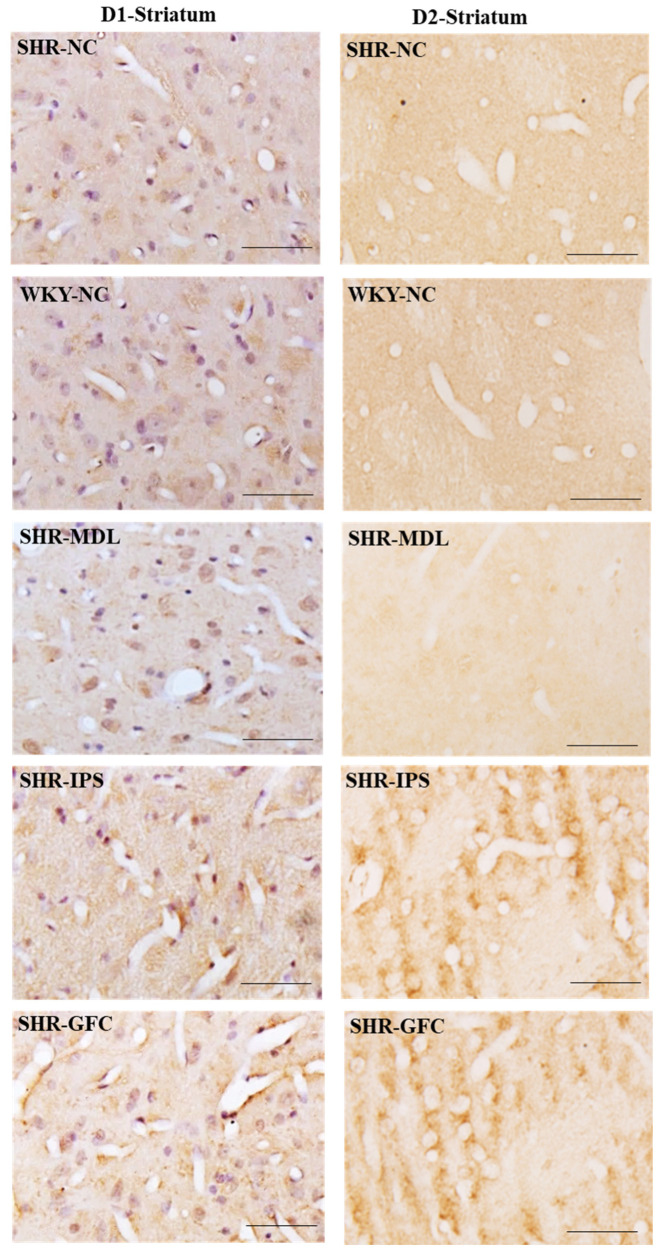
Photomicrographs of striatal sections immunostained for DA-D1 (**left** panel) and DA-D2 (**right** panel) receptors. Note the expression pattern DA-D1 and DA-D2 receptors in different groups which agrees with mean gray intensity of immunoblot of Western blot analysis shown in Figure 9C,D. Scale bar = 50 µm.

**Figure 12 ijms-25-02300-f012:**
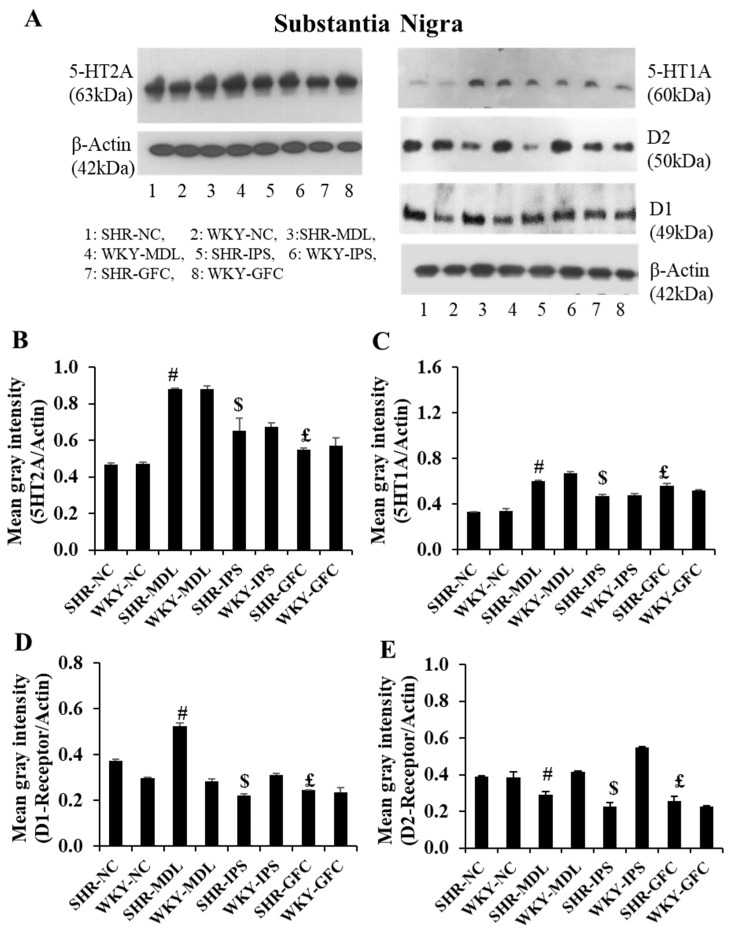
(**A**): Immunoblots of 5-HT2A, 5-HT1A, DA-D1, and DA-D2 receptors in the substantia nigra tissues. (**B**–**E**): Mean gray intensity of 5-HT2A (**B**) and 5-HT1A (**C**), DA-D1 (**D**), and DA-D2 (**E**) receptor immunobands normalized to actin band gray intensity. # SHR-NC vs. SHR-MDL, *p* < 0.05; $ SHR-NC vs. SHR-IPS, *p* < 0.05; £ SHR-NC vs. SHR-GFC, *p* < 0.05.

**Figure 13 ijms-25-02300-f013:**
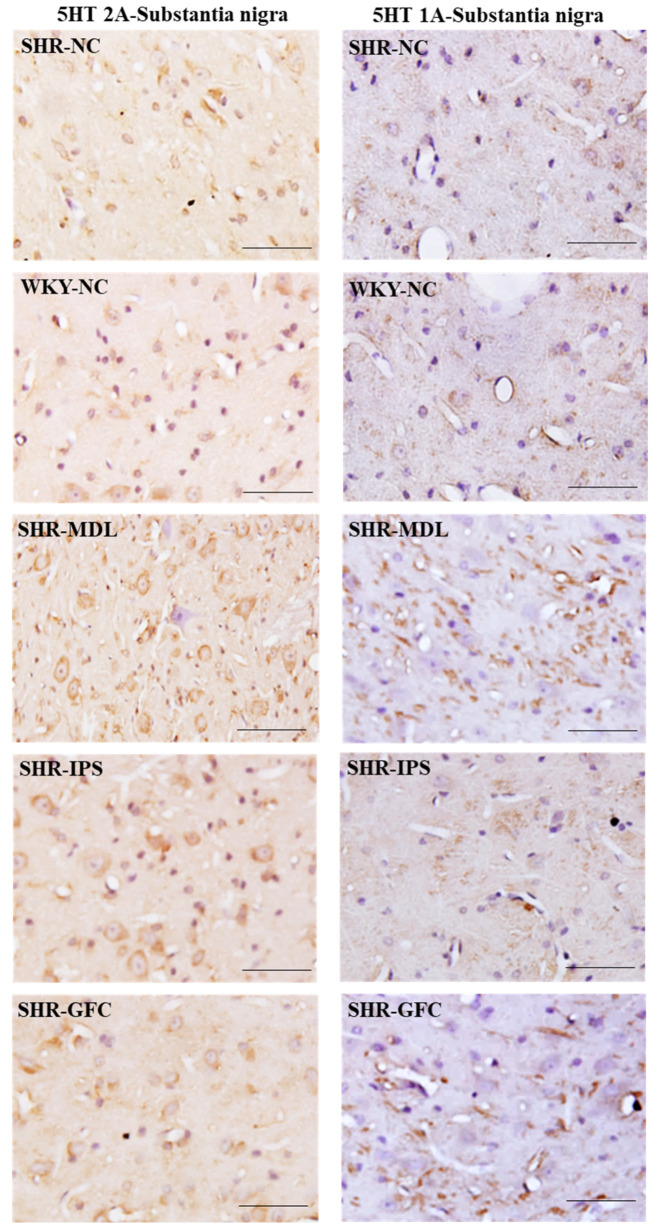
Photomicrographs of substantia nigra sections immunostained for 5-HT2A (**left** panel) and 5-HT1A (**right** panel) receptors. Note the expression pattern of 5-HT2A and 5-HT1A receptors in different groups which agrees with mean gray intensity of immunoblot of Western blot analysis shown in Figure 12B,C. Scale bar = 50 µm.

**Figure 14 ijms-25-02300-f014:**
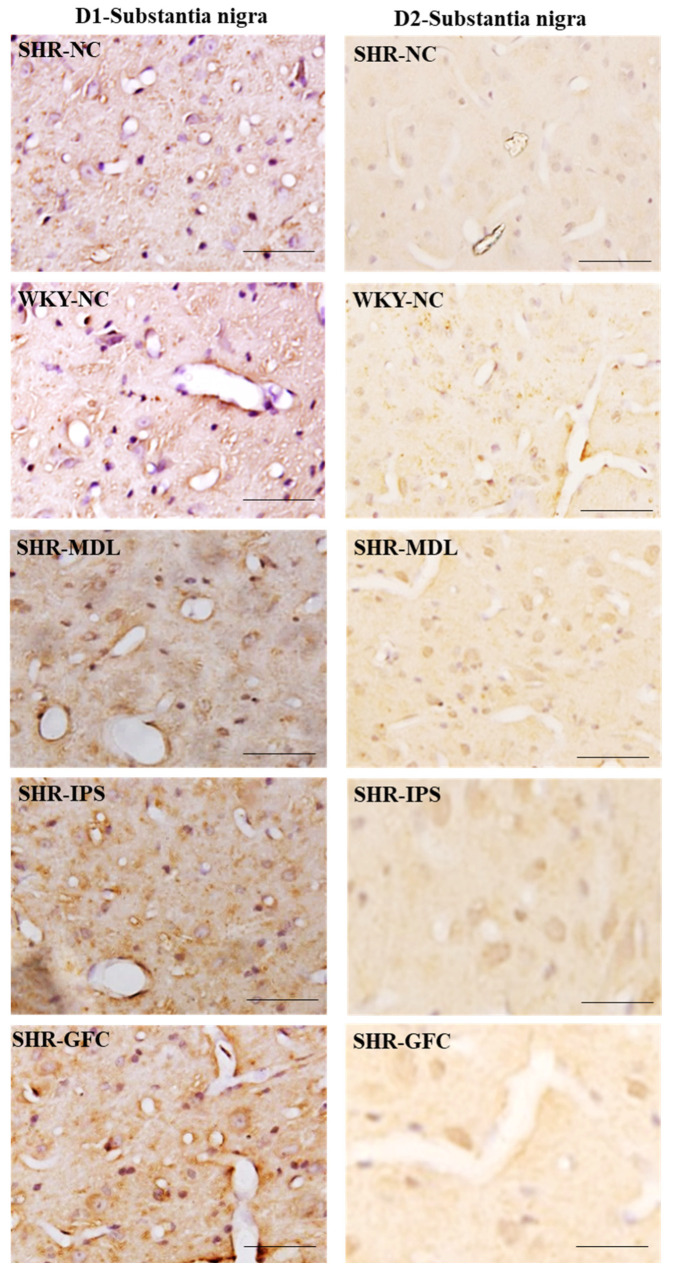
Photomicrographs of substantia nigra sections immunostained for DA-D1 (**left** panel) and DA-D2 (**right** panel) receptors. Note the expression pattern DA-D1 and DA-D2 receptors in different groups which agrees with mean gray intensity of immunoblot of Western blot analysis shown in Figure 12C,D. Scale bar = 50 µm.

## Data Availability

The data that support the findings of this study are available from the corresponding author upon reasonable request.

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
