# Peer review of "Serotonergic and Adrenergic Neuroreceptor Manipulation Ameliorates Core Symptoms of ADHD through Modulating Dopaminergic Receptors in Spontaneously Hypertensive Rats"

_ijms, 2024, doi:10.3390/ijms25042300_

Round 1
Reviewer 1 Report
Comments and Suggestions for Authors
It is an interesting topic which is rarely addressed.
However, there are many aspects that need to be improved:
Lines 11-12: ,,Therapeutic strategies mainly involves drugs which enhance DA and NE neurotransmissions”.
Instead of "drugs that enhance DA and NE neurotransmission," you can specify types of drugs or examples to make the text more concrete.
Lines 15-16: ,,It is not clear how does serotonergic receptors minimize the symptoms of ADHD by modulating the DA system”.
It should be reformulated to highlight the ambiguity more specifically.
Lines 31-32: ,,The study demonstrates that 5-HT1A agonist or 5-HT2A antagonist monotherapy alone can minimize the ADHD behaviour through modulating DA receptors”.
I think it can be adjusted to provide a more concrete conclusion or to emphasize the limitations of the study, if applicable.
Lines 40-42: ,,The disorder is highly complex due to multifactorial etiology and highly variable manifestations among the affected individuals. Although behavioral intervention is primary to minimize the ADHD symptoms, most patients need medications”.
You can provide a brief mention of various etiological factors of ADHD and the variability of manifestations among affected individuals to emphasize the complexity of this disorder.
Lines 45-46: ,,Apart from these two neurotransmitters the serotonin (5-HT) is also known to involve in ADHD “.
It would be beneficial if you could provide a brief explanation of how 5-HT is involved or provide references to support this statement.
Lines 52-54: ,,The mechanism of action of methylphenidate is by inhibiting DA and NE transporters and agonists activity of 5-HT1A neuroreceptor”.
Can you provide a brief explanation of the exact mechanism by which methylphenidate works by inhibiting the activity of DA and NE transporters and 5-HT1A agonists.
Lines 62-64: ,,Of late concurrent usage of stimulant and antipsychotic has been rationalized by suggesting that they likely interact with different receptor subtypes and do so in different pathways of the brain”.
Detail more about the reasons behind the argumentation of the simultaneous use of stimulants and antipsychotics to emphasize the importance of this strategy.
Lines 68-70: ,,However, analyses of twelve placebo-controlled trials in children and adolescents showed a greater risk of suicidal ideation during treatment with atomoxetine”.
Mention other possible side effects of atomoxetine, along with the increased risk of suicidal ideation, to provide a more balanced perspective on treatment.
Lines 77-79: ,,There were several case reports of sudden death in children taking a combination of clonidine and methylpheni date”.
Regarding the cases of sudden death in children who have taken a combination of clonidine and methylphenidate, can you specify if there is a certain age limit at which this risk is more pronounced.
Lines 88-89: ,,Risperidone being a relatively new generation of medication, its potential benefits and adverse effects in long term use and comorbid conditions is yet to be explored”.
You could add a few words about the available data on long-term use and possible comorbid conditions related to risperidone to give a more comprehensive perspective.
Lines 128-131: ,,The currently approved DA agonists are mainly D1/D2/D3 agonists which produces side-effects typical of D2/D3 agonists, namely a tendency to increase addictive behaviors like gambling, compulsive shopping, and hypersexuality”.
When referring to the side effects of DA agonists, you can provide specific examples of those side effects to better highlight potential problems.
Lines 136-138: ,,Since Alpha-2 adrenergic agonists is well known therapeutic component of ADHD treatment the experiment also have animal group who receive alpha-2 adrenergic agonists”.
When you mention the groups of animals receiving alpha 2 adrenergic agonists, you can provide more details about these groups, such as the type of agonists and the doses administered.
Lines 141-145: ,,Understanding the relationship of dopamine receptors with ADHD will help us to elucidate different roles of these receptors and to develop therapeutic approaches of ADHD. The results of the study are likely to throw more insights into the role of these neuroreceptors in the brain regions concerned with ADHD”.
In the final sentences, you can further clarify the specific purpose of the study and how the results might contribute to the understanding of ADHD and the development of therapeutic approaches.
Lines 148-150: ,,The mean distance travelled, and mean movement velocity was significantly high (p<0.001) in SHR rats who did not receive any treatment (SHR-NC) compared to their counterparts WKY rats who also did not receive any treatment (WKY-NC)”.
I think you can divide this idea into two separate sentences to improve clarity.
Lines 150-151: ,,SHR rats who received either MDL or IPS or GFC has shown a highly significant ..”
I think you can specify more clearly which group received each treatment to avoid ambiguity.
Lines 185-186: ” ..decrease in these study parameters when compared to SHR control rats in the study ….”
It would be better to replace phrases such as "these study parameters" with specific terms such as "average movement number" and "movement time," to make the text more accessible to readers.
Lines 190-191: ,,There was no significant difference noticed between MDL and IPS groups in these parameters”.
You should mention in the text that no significant difference was observed between the MDL and IPS groups. I think you can provide a brief explanation of the relevance of this finding and possible interpretations.
Lines 211-212: ,,There was no significant difference between MDL and IPS treated groups with regards to these parameters”.
I think you can give a brief explanation regarding the relevance of this finding and the possible interpretations.
I think you can provide a brief explanation of the relevance of this finding and possible interpretations.
Lines 266-267: ,,Expression of neuroreceptors (5-HT2A, 5-HT1A, DA-D1 and DA-D2) in prefrontal 266 cortex, striatum…”
It would be better to separate subsections referring to specific receptors (5-HT2A, 5-HT1A, DA-D1, DA-D2) to make the information easier to follow.
Lines 273-274: “In the striatum the receptor alteration was 273 more uniform in the ventral and dorsal parts….”
You can add a brief discussion of the clinical significance of the observed changes in receptor expression, particularly in the context of ADHD-type disorders.
Lines 607-609: ,,One of the limitations of our study is that we did not measure the extracellular DA, NE and 5-HT levels in the areas of the brain affected in ADHD”.
I appreciate that you present the limits of the study that you did not measure the extracellular levels of DA, NE and 5-HT in the areas affected by ADHD.
The references are not placed correctly: the publication year of the magazine is not bold-21,47, other references are underlined-32,33,34,38,40,41,45,48,49,55,67,68 or others are not written according to template-20,44,77,82.
My comments are only intended to make the paper better. Good luck!
Author Response
Lines 11-12: ,,Therapeutic strategies mainly involves drugs which enhance DA and NE neurotransmissions”.
Instead of "drugs that enhance DA and NE neurotransmission," you can specify types of drugs or examples to make the text more concrete.
Agreed and the text is revised
Lines 15-16: ,,It is not clear how does serotonergic receptors minimize the symptoms of ADHD by modulating the DA system”.
It should be reformulated to highlight the ambiguity more specifically.
The sentence is reformulated
Lines 31-32: ,,The study demonstrates that 5-HT1A agonist or 5-HT2A antagonist monotherapy alone can minimize the ADHD behaviour through modulating DA receptors”.
I think it can be adjusted to provide a more concrete conclusion or to emphasize the limitations of the study, if applicable.
Yes, the sentence is reformulated saying that in this animal model study the 5-HT1 agonist or 5-HT2A antagonist monotherapies were able to curtail the ADHD symptoms by differentially expressing DA receptors in different regions of the brain.
Lines 40-42: ,,The disorder is highly complex due to multifactorial etiology and highly variable manifestations among the affected individuals. Although behavioral intervention is primary to minimize the ADHD symptoms, most patients need medications”.
You can provide a brief mention of various etiological factors of ADHD and the variability of manifestations among affected individuals to emphasize the complexity of this disorder.
Agreed, the various etiological factors of ADHD and variability of manifestations and on this basis the importance of personalized pharmacological treatment option were added in the manuscript.
Lines 45-46: ,,Apart from these two neurotransmitters the serotonin (5-HT) is also known to involve in ADHD “.
It would be beneficial if you could provide a brief explanation of how 5-HT is involved or provide references to support this statement.
Agreed, the role of 5-HT in inhibitory response control and sustained attention were cited
Lines 52-54: ,,The mechanism of action of methylphenidate is by inhibiting DA and NE transporters and agonists activity of 5-HT1A neuroreceptor”.
Can you provide a brief explanation of the exact mechanism by which methylphenidate works by inhibiting the activity of DA and NE transporters and 5-HT1A agonists.
Methylphenidate inhibits the NE and DA transporters and thereby increases the level of NE and DA especially in the prefrontal cortex (in the synaptic cleft and making it available for neurotransmission) which is involved in ADHD. This has been added into the manuscript.
Lines 62-64: ,,Of late concurrent usage of stimulant and antipsychotic has been rationalized by suggesting that they likely interact with different receptor subtypes and do so in different pathways of the brain”.
Detail more about the reasons behind the argumentation of the simultaneous use of stimulants and antipsychotics to emphasize the importance of this strategy.
How stimulant and antipsychotics act differently in different regions of the brain and the rationale of using combination of drugs are highlighted in the manuscript
Lines 68-70: ,,However, analyses of twelve placebo-controlled trials in children and adolescents showed a greater risk of suicidal ideation during treatment with atomoxetine”.
Mention other possible side effects of atomoxetine, along with the increased risk of suicidal ideation, to provide a more balanced perspective on treatment.
The most common insignificant and less common but significant adverse effects of atomoxetine were added to the manuscript.
Lines 77-79: ,,There were several case reports of sudden death in children taking a combination of clonidine and methylphenidate”.
Regarding the cases of sudden death in children who have taken a combination of clonidine and methylphenidate, can you specify if there is a certain age limit at which this risk is more pronounced.
A case report claiming four cases of sudden death of children aged between 8 to 10 were added to the manuscript.
Lines 88-89: ,,Risperidone being a relatively new generation of medication, its potential benefits and adverse effects in long term use and comorbid conditions is yet to be explored”.
You could add a few words about the available data on long-term use and possible comorbid conditions related to risperidone to give a more comprehensive perspective.
Risperidone is used 'off-label' in several psychiatric conditions including ADHD. The available literature regarding its adverse effects in long-term treatment are few and they were added to the manuscript.
Lines 128-131: ,,The currently approved DA agonists are mainly D1/D2/D3 agonists which produces side-effects typical of D2/D3 agonists, namely a tendency to increase addictive behaviors like gambling, compulsive shopping, and hypersexuality”.
When referring to the side effects of DA agonists, you can provide specific examples of those side effects to better highlight potential problems.
Dopamine agonists were used in variety of neuropsychiatric disorder but it always warrants close monitoring of the patient as it can cause choreiform and dystonic movements. Hence formulation of dopamine agonist as a part of ADHD medication for children raises concerns for the possible extrapyramidal symptoms. It would be interesting to evaluate the effect of serotonergic manipulation on dopamine receptors and whether it would be beneficial in curtailing the ADHD symptoms.
Lines 136-138: ,,Since Alpha-2 adrenergic agonists is well known therapeutic component of ADHD treatment the experiment also have animal group who receive alpha-2 adrenergic agonists”.
When you mention the groups of animals receiving alpha 2 adrenergic agonists, you can provide more details about these groups, such as the type of agonists and the doses administered.
Included
Lines 141-145: ,,Understanding the relationship of dopamine receptors with ADHD will help us to elucidate different roles of these receptors and to develop therapeutic approaches of ADHD. The results of the study are likely to throw more insights into the role of these neuroreceptors in the brain regions concerned with ADHD”.
In the final sentences, you can further clarify the specific purpose of the study and how the results might contribute to the understanding of ADHD and the development of therapeutic approaches.
Included
Lines 148-150: ,,The mean distance travelled, and mean movement velocity was significantly high (p<0.001) in SHR rats who did not receive any treatment (SHR-NC) compared to their counterparts WKY rats who also did not receive any treatment (WKY-NC)”.
I think you can divide this idea into two separate sentences to improve clarity.
The interpretation of this result is mentioned, and it is expected to give clear idea to the readers.
Lines 150-151: ,,SHR rats who received either MDL or IPS or GFC has shown a highly significant ..”
I think you can specify more clearly which group received each treatment to avoid ambiguity.
Yes, in since in this journal format the methods section appears at the end, the readers need to be familiar with abbreviation used. Hence in this part of the result section the abbreviation used were highlighted in parentheses.
Lines 185-186: ” ..decrease in these study parameters when compared to SHR control rats in the study ….”
It would be better to replace phrases such as "these study parameters" with specific terms such as "average movement number" and "movement time," to make the text more accessible to readers.
Included throughout
Lines 190-191: ,,There was no significant difference noticed between MDL and IPS groups in these parameters”.
You should mention in the text that no significant difference was observed between the MDL and IPS groups. I think you can provide a brief explanation of the relevance of this finding and possible interpretations.
Included
Lines 211-212: ,,There was no significant difference between MDL and IPS treated groups with regards to these parameters”.
I think you can give a brief explanation regarding the relevance of this finding and the possible interpretations.
Included
Lines 266-267: ,,Expression of neuroreceptors (5-HT2A, 5-HT1A, DA-D1 and DA-D2) in prefrontal 266 cortex, striatum…”
It would be better to separate subsections referring to specific receptors (5-HT2A, 5-HT1A, DA-D1, DA-D2) to make the information easier to follow.
Expression of each neuroreceptors in different brain regions were separately discussed in the result section. Do you want to show each receptors in different regions (instead of different regions where each receptor expression was discussed?)
Lines 273-274: “In the striatum the receptor alteration was 273 more uniform in the ventral and dorsal parts….”
You can add a brief discussion of the clinical significance of the observed changes in receptor expression, particularly in the context of ADHD-type disorders.
Included
Lines 607-609: ,,One of the limitations of our study is that we did not measure the extracellular DA, NE and 5-HT levels in the areas of the brain affected in ADHD”.
I appreciate that you present the limits of the study that you did not measure the extracellular levels of DA, NE and 5-HT in the areas affected by ADHD.
Thanks, hope we should be able to measure the brain amines in near future
The references are not placed correctly: the publication year of the magazine is not bold-21,47, other references are underlined-32,33,34,38,40,41,45,48,49,55,67,68 or others are not written according to template-20,44,77,82.
It is rectified
My comments are only intended to make the paper better. Good luck!
Thanks once again for the wonderful review and inputs!
Reviewer 2 Report
Comments and Suggestions for Authors
The manuscript ‘Serotonergic and adrenergic neuro receptor manipulation ameliorate core symptoms of ADHD through modulating dopaminergic receptors in spontaneously hypertensive rats‘ aims to explore the effectiveness of manipulating serotonergic and alpha-2 adrenergic receptors in addressing the core symptoms of ADHD. Additionally, it seeks to elucidate the impact of such manipulation on dopamine (DA) neuroreceptors within specific brain regions associated with ADHD in 5-HT1A agonist and 5-HT2A antagonist spontaneously hypertensive rats (SHR). The study illustrates that employing either a 5-HT1A agonist or a 5-HT2A antagonist as a standalone therapeutic approach is effective in reducing ADHD-like behaviors, achieving this outcome through the modulation of dopamine (DA) receptors.
The study is nicely conducted and the results are promising. After going through the manuscript, I have following comment for the author.
1. It is important to note that findings in animal models don't always directly translate to humans, and the complex interplay between neurotransmitter systems makes it challenging to draw straightforward conclusions. I would suggest the authors to briefly discuss this point in the discussion.
2. Please provide the conclusion of the study.
Comments on the Quality of English LanguageMinor grammatical and syntax correction needed.
Author Response
The manuscript ‘Serotonergic and adrenergic neuro receptor manipulation ameliorate core symptoms of ADHD through modulating dopaminergic receptors in spontaneously hypertensive rats‘ aims to explore the effectiveness of manipulating serotonergic and alpha-2 adrenergic receptors in addressing the core symptoms of ADHD. Additionally, it seeks to elucidate the impact of such manipulation on dopamine (DA) neuroreceptors within specific brain regions associated with ADHD in 5-HT1A agonist and 5-HT2A antagonist spontaneously hypertensive rats (SHR). The study illustrates that employing either a 5-HT1A agonist or a 5-HT2A antagonist as a standalone therapeutic approach is effective in reducing ADHD-like behaviors, achieving this outcome through the modulation of dopamine (DA) receptors.
The study is nicely conducted, and the results are promising. After going through the manuscript, I have following comment for the author.
- It is important to note that findings in animal models don't always directly translate to humans, and the complex interplay between neurotransmitter systems makes it challenging to draw straightforward conclusions. I would suggest the authors to briefly discuss this point in the discussion.
Fully agree with the reviewer and the same is included in the manuscript.
- Please provide the conclusion of the study.
We conclude that the 5-HT1A agonists and 5-HT2A antagonists monotherapies alone could reduce the core symptoms of ADHD by altering the expression of DA1 and DA2 neuroreceptors in prefrontal cortex, striatum and substantia nigra.
